# CoKV: Optimizing LLM Inference with Game-Theoretic Adaptive KV Cache

## Abstract

Large language models (LLMs) have achieved remarkable success in various aspects of human life. However, one of the major challenges in deploying these models is the substantial memory consumption required to store key-value pairs (KV), which imposes significant resource demands. Recent research has focused on KV cache budget allocation, with several approaches proposing head-level budget distribution by evaluating the importance of individual attention heads. These methods, however, assess the importance of heads independently, overlooking their cooperative contributions within the model, which may result in a deviation from their true impact on model performance. In light of this limitation, we propose CoKV, a novel method that models the cooperation between heads in model inference as a cooperative game. By attributing the contribution of each head within the model, CoKV can more effectively allocate the cache budget in KV cache techniques such as eviction and quantization. Extensive experiments demonstrate the effectiveness of CoKV on long-context benchmarks (e.g., LongBench, NIAH, and RULER) and mathematical reasoning benchmarks (e.g., GSM8K and MATH) across multiple model families, including Qwen, Llama, and Mistral. Code is provided in `https://anonymous.4open.science/r/CoKV-40AC`.

## 1 Introduction

Large language models (LLMs) are widely applied across various domains, including content generation (Li et al., 2024a), automated services (Chen et al., 2024a), and decision support systems (Hager et al., 2024). With the widespread application of large language models (LLMs), reducing the cost of inference services has become increasingly important. LLMs consist of multiple transformer blocks that store key and value states (KV) during inference. KV cache allows efficient decoding in token generation without recomputing key and value states by using previously cached KV pairs. However, the KV cache becomes excessively large when processing long sequences or a large number of inputs, inevitably straining GPU memory, thereby substantially raising deployment costs and hardware requirements for large-scale applications.

To address this challenge, research efforts have advanced on several fronts. Some studies have explored methods for ranking the importance of tokens within a single attention head, retaining only the top $k$ most significant ones. For example, H2O (Zhang et al., 2023b) evaluates token importance using the sum of attention weights. StreamingLLM (Xiao et al., 2024) directly removes KV from the middle segment of the cache to reduce the cache size as they incorporate less information. SnapKV (Li et al., 2024b) calculates token scores by pooling the attention weights between tokens in the local window and those in the cache. In parallel, several studies have also investigated strategies for optimizing KV quantization to reduce KV cache costs, such as Kvquant (Hooper et al., 2024) and OTT (Su et al., 2025). Recently, some studies have recognized that the importance of each attention head varies, enabling methods like AdaKV (Feng et al., 2025), HeadKV (Fu et al., 2025) and DuoAttention (Xiao et al., 2025b). AdaKV improves budget utilization by adaptively allocating the overall budget across different attention heads based on their varied concentration degrees. HeadKV evaluates the retrieval-reasoning scores of different heads and allocates a larger cache size to those with higher scores. DuoAttention uses a reinforcement learning-based algorithm with synthetic data to identify retrieval heads.

While prior work on head importance evaluation has made significant advancements in adaptive KV cache management, we observe that several challenges remain unresolved in current approaches. Some existing methods evaluate attention head importance independently. For example, AdaKV evaluates the concentration degrees of heads while HeadKV assesses the retrieval-reasoning capability of each head in isolation as a measure of importance. However, these approaches treat heads as isolated units, overlooking the fact that their true importance emerges from their cooperation rather than individual capabilities. As a result, independently assessing head importance may lead to suboptimal allocation. DuoAttention frames each attention head as an agent within a reinforcement learning framework, thereby incorporating interactions among heads. However, this approach suffers from unstable policy convergence, which can hinder its practical application. Based on these insights, we propose **CoKV** (**Co**operation-based **K**ey-**V**alue Cache), a method that evaluates the contribution of all attention heads in their cooperation within the model based on game-theoretic utilities and dynamically allocates cache budgets based on their contribution.

CoKV is inspired by the Shapley value (Shapley, 1953), a seminal concept in cooperative game theory that offers a mathematically rigorous framework for fair contribution allocation. The Shapley value of a player $p_i$ measures the expected marginal contribution that $p_i$ provides to a coalition of players. In this work, each attention head can be treated as a player, with its importance assessed via its Shapley value. The marginal contribution is defined as $\mathcal{U}(\mathcal{S} \cup \{p_i\}) - \mathcal{U}(\mathcal{S})$ where $\mathcal{S}$ is a coalition of players excluding $i$ and $\mathcal{U}$ is the utility function. A simple intuition for computing the Shapley value of each head in LLMs is to define $\mathcal{U}$ as the model performance metric. This is a #P-hard (Deng & Papadimitriou, 1994) problem as there are an exponential number of coalitions and corresponding marginal contributions, thus requiring an enormous number of model inferences. Although many studies (Jia et al., 2019; Mitchell et al., 2022) have explored approximating the Shapley value to reduce computational costs, the process of applying these methods to evaluate the importance of heads in LLMs remains prohibitively expensive.

The computational bottleneck in calculating the Shapley value arises from the fact that each sample of the marginal contribution can only be applied to a single player. Fortunately, Shapley value can be expressed as the expectation of the weighted complementary contribution, defined as $\mathcal{U}(\mathcal{S}) - \mathcal{U}(\mathcal{N} \setminus \mathcal{S})$, where $\mathcal{N}$ represents the set of all players (Zhang et al., 2023a). Complementary contribution has an advantage over the marginal contribution in that $\mathcal{U}(\mathcal{S}) - \mathcal{U}(\mathcal{N} \setminus \mathcal{S})$ can be used to update the Shapley values for all players in $\mathcal{S}$. By expressing the Shapley value in terms of complementary contributions, we can interpret it as an expectation over these contributions computed at different coalition sizes $|\mathcal{S}|$. However, in the LLM setting, the cost of computing the complementary contributions in all coalition sizes is still prohibitively high. We observe that the average complementary contribution of a single player at different coalition sizes exhibits a strong correlation in Appendix Section E.4. This insight allows us to approximate attention head importance by computing complementary contributions at only a few selected coalition sizes, rather than evaluating all possible sizes (i.e., from 1 to $|\mathcal{N}|$). By focusing on a few representative coalition sizes, we can significantly reduce the computational cost of estimating the contributions of heads. Additionally, we provide a theoretical error bound of this approach and demonstrate its efficiency.

CoKV is a simple-yet-effective method and can integrate well with other inference optimization techniques. We integrate CoKV with widely used methods, including FlashAttention (Dao et al., 2022) and group query attention (GQA) (Ainslie et al., 2023). CoKV achieves state-of-the-art performance in LongBench (Bai et al., 2024) using Qwen3-32B, Llama-3-8B-Instruct (Dubey et al., 2024) and Mistral-7B Jiang et al. (2023) models. Results from the Llama-3-8B-Instruct model show that when each KV cache retains an average of 128 KV pairs ($1.6\%$ of the full cache), it achieves $97.29\%$ of the performance of the full KV cache. Furthermore, when each cache retains just 512 tokens on average, CoKV outperforms the full KV cache in terms of average accuracy. This demonstrates that CoKV not only reduces computational costs but also improves inference performance by identifying which heads benefit from cache retention and which may have a detrimental effect. For Qwen3-32B, CoKV achieve $98.83\%$ of the performance of the full KV when retains an average of 1024KV pairs($12.8\%$). Additionally, we evaluate all methods within the token range up to 61k in the Needle-in-a-Haystack test and the RULER dataset (Hsieh et al., 2024), which are widely recognized benchmarks for evaluating long-text processing capabilities of LLMs, where CoKV also demonstrated the best performance. Experiments on mathematical reasoning datasets also demonstrate that CoKV possesses strong cross-task capabilities.

## 2 PRELIMINARIES

In this section, we first formalize the key-value caching and compression mechanism in multi-head attention. We then present the Shapley value framework as a principled approach for quantifying the importance of individual attention heads.

### 2.1 KEY-VALUE CACHING AND COMPRESSION

In Multi-Head Attention (MHA), for each attention head $h_i$ in one layer, the embedded input $X = \{x_1, x_2, \ldots, x_m\} \in \mathbb{R}^{m \times d_{\text{model}}}$ of $m$ tokens is mapped into different subspaces using query $W_i^Q$, key $W_i^K$, and value $W_i^V \in \mathbb{R}^{d_{\text{model}} \times d_h}$ matrices:

$$Q_i = XW_i^Q, K_i = XW_i^K, V_i = XW_i^V \in \mathbb{R}^{m \times d_h}$$

where $d_h$ is the dimension of attention heads, $d_h = d/\tau$, and $\tau$ is the number of heads in one layer.

All the computed $KV$ for the input sequence are cached to avoid recalculating them during the subsequent decoding stages. Assume there is a new input token $x \in \mathbb{R}^{1 \times d_{\text{model}}}$, then it will be mapped to a new query, key, and value as follows,

$$q_i = xW_i^Q, k_i = xW_i^K, v_i = xW_i^V \in \mathbb{R}^{1 \times d_h}.$$

The KV cache is updated by adding the new key and value pair

$$K_i = \text{Cat}[K_i, k_i], V_h = \text{Cat}[V_i, v_i].$$

The attention output is computed as follows $O_i = A_i V_i$ where $A_i = \text{softmax}(q_i K_i^T / \sqrt{d_h})$. The final output $y \in \mathbb{R}^{1 \times d_{\text{model}}}$ is obtained through a linear transformation

$$y = \text{Cat}[O_1, \cdots, O_\tau]W^O$$

where $W^O \in \mathbb{R}^{d \times d_{\text{model}}}$ output weight matrix.

Due to space limitations, we present the introduction of KV cache eviction and KV cache quantization in Appendix Section C.

### 2.2 SHAPLEY VALUE

Consider a set of players $\mathcal{N} = \{p_1, \ldots, p_n\}$. A *coalition* $\mathcal{S}$ is a subset of $\mathcal{N}$ that cooperates to complete a task. A utility function $\mathcal{U}(\mathcal{S})$ ($\mathcal{S} \subseteq \mathcal{N}$) is the utility of coalition $\mathcal{S}$ for the task. The marginal contribution of player $p_i$ with respect to a coalition $\mathcal{S}$ is $\mathcal{U}(\mathcal{S} \cup \{p_i\}) - \mathcal{U}(\mathcal{S})$. The Shapley value measures the expectation of marginal contribution of player $p_i$ in all possible coalitions. That is

$$\mathcal{SV}_i = \frac{1}{n} \sum_{\mathcal{S} \subseteq \mathcal{N} \setminus \{p_i\}} \frac{\mathcal{U}(\mathcal{S} \cup \{p_i\}) - \mathcal{U}(\mathcal{S})}{\binom{n-1}{|\mathcal{S}|}}. \quad (1)$$

According to Equation 1, it is evident that computing the exact Shapley value requires enumerating the utilities for all possible subsets of players and each marginal contribution can only be used to update the Shapley value of a single player. Therefore, the computational complexity of exactly calculating the Shapley value is exponential. Recently, the Shapley value of player $p_i$ is proven to be equal to the weighted complementary contributions (Zhang et al., 2023a) as follows,

$$\mathcal{SV}_i = \frac{1}{n} \sum_{\mathcal{S} \subseteq \mathcal{N} \setminus \{p_i\}} \frac{\mathcal{U}(\mathcal{S}) - \mathcal{U}(\mathcal{N} \setminus \mathcal{S})}{\binom{n-1}{|\mathcal{S}|}}. \quad (2)$$

$\mathcal{U}(\mathcal{S}) - \mathcal{U}(\mathcal{N} \setminus \mathcal{S})$ is called complementary contribution which has an advantage that can be reused to update Shapley value estimation for all players in $\mathcal{S}$. In the context of KV caches, attention heads are treated as players for evaluating their importance to each specific task. $\mathcal{U}(\mathcal{S})$ is defined as the model accuracy when the attention heads in $\mathcal{N} \setminus \mathcal{S}$ are masked, we retain only the KV pairs within the local window for masked heads.

# 3 IMPORTANCE-AWARE KV CACHE COMPRESSION VIA SLICED SHAPLEY VALUE

Our method consists of two phases. First, we precompute the importance scores for each attention head. Second, these scores are utilized for KV cache compression during inference. The overview of our approach is illustrated in Figure 1.

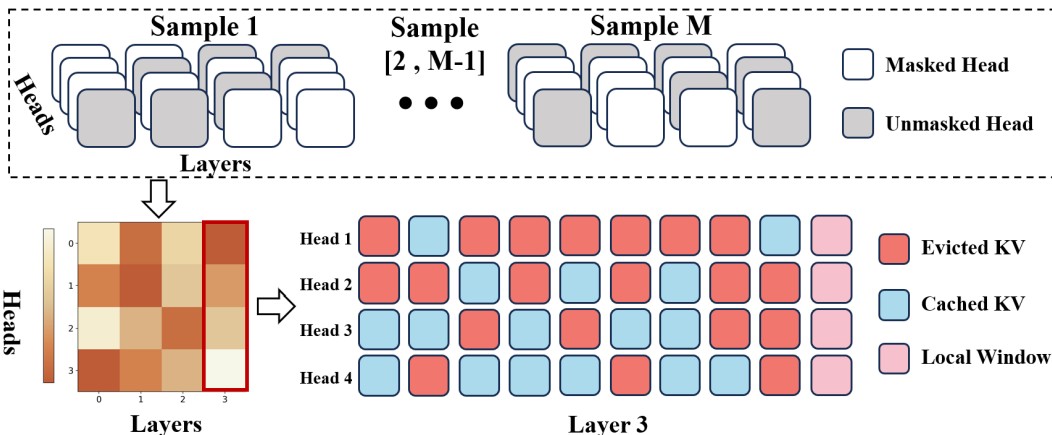

Figure 1: Overview of our proposed method: (1) **Head Importance Evaluation (Upper Part):** For a 4-layer × 4-head model, We measure head importance using the Sliced Shapley Value (SSV). To approximate SSV, we sample $M$ different sets of masked heads and compute their complementary contributions. The average complementary contribution of each head is its estimated SSV. (2) **KV Cache Compression (Lower Part):** Using the 4 heads in Layer 3 and the KV cache eviction method as an example, each head stores KV pairs for a small local window of recent tokens. Heads with higher SSV (represented by darker areas in the heatmap) are allocated more cache size to retain KV pairs prior to the local window. For adaptive KV cache quantization, we can assign heads with higher SSV more bits, while heads with lower SSV receive fewer bits.

## 3.1 HEAD IMPORTANCE EVALUATION

Although the complementary contribution helps in increasing efficiency when approximating the Shapley value, it is still computationally costly, especially in the LLM setting. Given a set of players $\mathcal{N} = \{p_1, \ldots, p_n\}$, a coalition of $j$ players $(1 \leq j \leq n)$ is called a *j-coalition*. Moreover, for a player $p_i$ $(1 \leq i \leq n)$, a $j$-coalition that contains $p_i$ is called a $(i,j)$-*coalition*. Denote by $\mathfrak{S}_{i,j} = \{\mathcal{S} \cup \{p_i\} | \mathcal{S} \subseteq \mathcal{N} \setminus \{p_i\}, |\mathcal{S}| = j - 1\}$ the set of $(i,j)$-coalitions, and by $\mathcal{SV}_{i,j}$ *the expected complementary contributions of* $(i,j)$-coalitions. That is,

$$\mathcal{SV}_{i,j} = \sum_{\mathcal{S} \in \mathfrak{S}_{i,j}} \frac{\mathcal{U}(\mathcal{S}) - \mathcal{U}(\mathcal{N} \setminus \mathcal{S})}{\binom{n-1}{j-1}}. \tag{3}$$

It is clear that $\mathcal{SV}_i = \frac{1}{n} \sum_{j=1}^{n} \mathcal{SV}_{i,j}$. Computing the Shapley value needs to calculate $\mathcal{SV}_{i,j}$ for $j$ ranging from 1 to $n$, which becomes costly when $n$ is large.

We observe that the expected complementary contributions of $j$-coalitions for heads in LLMs follow a similar distribution across different $j$ values, as shown in Appendix Section E.4. This suggests that the contributions of heads can be effectively captured using a subset of $j$-coalitions. Based on this insight, we propose assessing the importance of heads using the expected complementary contribution of several $j$-coalitions, which can significantly reduce the computation cost while maintaining effectiveness. Formally, we introduce a new definition called the Sliced Shapley value .

**Definition 1 (Sliced Shapley Value)** Let $\mathcal{H} \subseteq \{1, \cdots, n\}$ denote the selected set of $j$-coalitions, representing a specific slice of the coalition size space. The *Sliced Shapley value* of head $h_i$ with

respect to $\mathcal{H}$ is defined as:

$$\mathcal{SSV}_i^{\mathcal{H}} = \frac{1}{|\mathcal{H}|} \sum_{j=1}^{n} \mathcal{SV}_{i,j} \cdot \mathbb{I}_j^{|\mathcal{H}|}, \tag{4}$$

where $\mathbb{I}_j^{\mathcal{H}}$ is an indicator function, which is 1 if $j$ is the element in $\mathcal{H}$ and 0 otherwise.

**Theorem 1** *Assume $\mathcal{SV}_{i,j} \in [a, b]$ for all $j$, and let $R = b - a$. Then, for any $\delta \in (0, 1)$, with probability at least $1 - \delta$, $|\mathcal{SV}_i - \mathcal{SSV}_i^{\mathcal{H}}| \leq R\sqrt{\frac{(n-|\mathcal{H}|+1)\ln(2/\delta)}{2|\mathcal{H}|n}}$. Furthermore, the established bound implies that the error $|\mathcal{SV}_i - \mathcal{SSV}_i^{\mathcal{H}}|$ is $\mathcal{O}\left(\sqrt{1/|\mathcal{H}|}\right)$. The proof is provided in Appendi Section D.1.*

---

**Algorithm 1:** Evaluating Head Importance in LLMs.

**input** :Heads $\mathcal{N} = \{h_1, \ldots, h_n\}$ and sampling number $\mathcal{M} > 0$
**output** :approximate Sliced Shapley value $\overline{\mathcal{SSV}_i^{\mathcal{H}}}$ for each head $h_i$ $(1 \leq i \leq n)$

1   $\overline{\mathcal{SV}_i^{\mathcal{H}}} \leftarrow 0$ $(1 \leq i \leq n)$; $\overline{\mathcal{SV}_{i,j}}, m_{i,j} \leftarrow 0$ $(1 \leq i, j \leq n)$;
2   **for** *k=1 to $\mathcal{M}$* **do**
3      let $\pi^k$ be a random permutation of $\{1, \ldots, n\}$;
4      let $i$ be a randomly selected element from the set $\mathcal{H}$;
5      $\mathcal{S} \leftarrow \{\pi^k(1), \ldots, \pi^k(i)\}$;
6      $\mathcal{N} \setminus \mathcal{S} \leftarrow \{\pi^k(i+1), \ldots, \pi^k(n)\}$;
      // $\mathcal{U}(\mathcal{S})$ is the model performance when heads in $\mathcal{N} \setminus \mathcal{S}$ are masked and vice versa for $\mathcal{U}(\mathcal{N} \setminus \mathcal{S})$.
7      $u \leftarrow \mathcal{U}(\mathcal{S}) - \mathcal{U}(\mathcal{N} \setminus \mathcal{S})$;
8      **for** *j=1 to i* **do**
9         $\overline{\mathcal{SV}_{\pi^k(j),i}} += u$;
10        $m_{\pi^k(j),i} += 1$;

11   **for** *i = 1 to n* **do**
12      $\overline{\mathcal{SSV}_i^{\mathcal{H}}} = \frac{1}{\mathcal{H}} \sum_{j=1}^{n} \overline{\mathcal{SV}_{i,j}} / m_{i,j}$;
13   **return** $\overline{\mathcal{SSV}_1^{\mathcal{H}}}, \ldots, \overline{\mathcal{SSV}_n^{\mathcal{H}}}$.

---

**Algorithm Description.** The detailed steps of approximating $\mathcal{SSV}_i^{\mathcal{H}}$ are shown in Algorithm 1. In each iteration, sample a random permutation $\pi^k$ of the heads $\{h_1, \ldots, h_n\}$, which defines a random ordering of the heads. Randomly select a split point and create a set $\mathcal{S}$ of selected heads. Mask heads in the set $\mathcal{N} \setminus \mathcal{S}$, and evaluate the model accuracy after masking, which is denoted as $\mathcal{U}(\mathcal{S})$. Similarly, calculate $\mathcal{U}(\mathcal{N} \setminus \mathcal{S})$ by masking heads in $\mathcal{S}$ (Lines 3-6). For each head in $\mathcal{S}$, update $\mathcal{SV}_{\pi^k(j),i}$ and count matrix $m_{\pi^k(j),i}$ (Lines 7-10). After $\mathcal{M}$ iterations are completed, calculate the approximated Sliced Shapley value for each head by averaging the complementary contributions.

**Theorem 2** *Algorithm 1 returns an $(\epsilon, \delta)$-approximation of Sliced Shapley value with time complexity $\mathcal{O}(\frac{T|\mathcal{H}|ln\frac{2|\mathcal{H}|}{\delta}}{\epsilon^2})$ where $T$ is the time cost of evaluating a complementary contribution which is the time to inference on the validation dataset of each task in our setting. In contrast, Shapley value requires the time complexity of $\mathcal{O}(\frac{Tnln\frac{2n}{\delta}}{\epsilon^2})$ to achieve an $(\epsilon, \delta)$-approximation. The proof is provided in Appendix Section D.2.*

## 3.2 KV Cache Compression

In this section, we present how our proposed head importance evaluation method can be effectively applied to KV cache compression. We demonstrate its application in two primary directions, KV cache eviction and KV cache quantization. While our main focus and contributions lie in the eviction-based approach, we also show that the sameimportance scores can be seamlessly integrated into quantization frameworks to achieve superior performance compared to existing methods.

---

**Algorithm 2:** Token Eviction Using CoKV.

---

**input** : Shared budget size $B$, local window size $s$, tokens in local window $X^{win} \in \mathbb{R}^{s \times d}$, KV in local window $\{K_i^{win}, V_i^{win}\}$, KV outside local window $\{K_i^{out}, V_i^{out}\}$

**output** : Retained KV Cache $\{\hat{K}_i, \hat{V}_i\}$

1 $Q_i^{win} = X^{win} W_i^Q$;
   // Compute attention weights of queries in local window and prefix Keys.

2 $\overline{A}_i = \text{softmax}(Q_i^{win} K_i^T)$;

3 $\overline{A}_i = \overline{A}_i.maxpooling(dim = 1).mean(dim = 0)$;
   // Calculate token scores outside the local window.

4 Get $c_i$ using Algorithm 1 and Equation 5;

5 $indices = \overline{A}_i.topk(c_i).indices$;

6 Select $\{\hat{K}_i, \hat{V}_i\}$ from $\{K_i^{out}, V_i^{out}\}$ according $indices$;

7 $\{\hat{K}_i, \hat{V}_i\} = \text{Cat}(\{\hat{K}_i, \hat{V}_i\}, \{K_i^{win}, V_i^{win}\})$;
   // Keep top $c_i$ KV pairs in the cache.

8 **return** Retained KV Cache $\{\hat{K}_i, \hat{V}_i\}$.

---

**KV Cache Eviction Budget Allocation.** An intuitive approach suggests that the least important heads, which contribute minimally or even negatively to the model performance, may not require cache allocation. Let $\alpha$ represent the number of such heads, which serves as the sole hyperparameter in our allocation scheme. For the remaining $n - \alpha$ heads, we employ a normalization method to normalize their importance scores and allocate the cache size proportionally based on their normalized scores.

Specifically, we normalize their contributions using min-max normalization for the $n - \alpha$ heads:

$$\mathcal{NSV}_i^{\mathcal{H}} = \frac{\mathcal{SSV}_i^{\mathcal{H}} - \min^{\alpha}(\mathcal{SSV}^{\mathcal{H}})}{\max(\mathcal{SSV}^{\mathcal{H}}) - \min^{\alpha}(\mathcal{SSV}^{\mathcal{H}})},$$

where $\min^{\alpha}(\cdot)$ and $\max(\cdot)$ extract the $\alpha$-th smallest and maximum value, respectively. For the $\alpha$ heads with the smallest Sliced Shapley values, we set the normalized score as 0. This ensures that all normalized scores lie in the range $[0, 1]$.

Next, the cache size $c_i$ allocated to head $h_i$ is determined by the local window size $s$ and linearly distributing the remaining shared cache size $B$ based on the normalized scores:

$$c_i = B \cdot \frac{\mathcal{NSV}_i^{\mathcal{H}}}{\sum_{j=1}^n \mathcal{NSV}_j^{\mathcal{H}}} + s. \tag{5}$$

**Algorithm Description.** The detailed KV cache eviction steps for a single head are outlined in Algorithm 2. First, we allocate the KV cache size for each head based on their normalized Sliced Shapley values. Next, we rank the importance of KV pairs within each head following SnapKV. Specifically, the most recent tokens within local windows guide the KV cache selection. Attention scores from these local windows to the remaining tokens are aggregated via pooling, with higher-scoring tokens retained in the cache for each head.

**Remark on KV Cache Quantization.** The head importance scores derived by our method are not limited to eviction and can be directly applied to guide non-uniform quantization strategies. Specifically, our scores enable an adaptive bit allocation scheme where more important heads are assigned higher precision. This principle can complement advanced quantization techniques like Kvquant (Hooper et al., 2024), allowing for a head-aware quantization policy that operates on top of their sophisticated per-channel methods.

In our experiments, we demonstrate that a simple integration, which allocates bits proportionally to our importance scores, consistently outperforms baselines that use alternative head importance metrics under the same average bit-width. This validates the general utility of our cooperation-based evaluation framework across different compression paradigms.

## 4 EXPERIMENTS

In this section, we present the evaluation results of our method on the LongBench benchmark. Due to space limitations, detailed results are provided in the appendix, including an ablation study on coalition sizes (Appendix Section E.2), the precomputation cost analysis of CoKV (Section E.3), evaluations on GSM8K and MATH (Section E.6), an analysis of CoKV's general ability (Section E.7, the Needle-in-a-Haystack test (Section E.8), and experiments on the RULER dataset (Section E.9).

### 4.1 EXPERIMENT SETTINGS

**Datasets.**  Details of all experimental datasets are provided in Appendix Section E.1.

**Baselines and Settings.**  We compare CoKV with four strong KV cache compression methods. All methods keep the same total cache size for fair comparison. Besides, we implement all methods with GQA Ainslie et al. (2023) and FlashAttention Dao et al. (2022) for efficient computation.

- **SnapKV** Li et al. (2024b) uses the last several tokens as local windows. Attention scores from these windows to the remaining tokens are pooled to guid the KV selection in each head.
- **PyramidKV** Cai et al. (2024) allocates more KV cache to lower layers to retain key information while reducing the budget for higher layers where information is already aggregated.
- **Ada-KV** Feng et al. (2025) dynamically allocates budgets to heads within each layer based on their concentration degrees, and can be combined with SnapKV or PyramidKV. Ada-SnapKV is used as the baseline due to its superior performance over Ada-PyramidKV.
- **HeadKV-R2** Fu et al. (2025) allocate budgets to heads based on their retrieval-reasoning score, and it uses SnapKV to rank the importance of KV pairs in each head.

We evaluate CoKV on the Qwen3-32B, Llama-3-8B-Instruct and Mistral-7B-Instruct-v0.2 models. Due to the page limit, the Mistral-7B-Instruct-v0.2 results are provided in Appendix. For test data that exceeds the maximum input length of Llama-3-8B-Instruct, we adopt the approach of HeadKV by utilizing the first 4k tokens and the last 4k tokens. Following standard practices in Ada and HeadKV, we perform cache eviction after the prefilling phase of each layer for consistent comparison. In GQA, a group of heads shares the same KV cache. We treat each cache within a group as a player in the cooperative game, evaluating their Sliced Shapley value to determine their importance scores. For HeadKV-R2, we calculate the importance score of each group by averaging the retrieval-reasoning scores of the heads within the group. This adaptation ensures compatibility with GQA, as HeadKV is implemented with MHA in the original paper. The context length for headkv score detection was configured to the maximum capacity of an H100 96G GPU for computing the head scores in Qwen-3-32B. For the Llama-3-8B-Instruct and Mistral-7B-Instruct-v0.2 models, we adopted the scores reported by the authors. In CoKV, we use for coalition sizes $\mathcal{H} = \{32, 64, 96, 128\}$ for Llama-3-8B-Instruct and Mistral-7B-Instruct-v0.2 which have 256 groups(32 layers, 8 groups in each layer), and only one coalition size $\mathcal{H} = \{256\}$ for Qwen3-32B (64 layers, 8 groups in each layer). Our ablation experiment of $\mathcal{H}$ show that CoKV with only one coalition size works well, but more slices will enhance CoKV. Following HeadKV-R2, we set the local window size to 8. We randomly split each dataset into a very small validation dataset and a test dataset. For Llama-3-8B-Instruct and Mistral-7B-Instruct-v0.2, we construct validation sets of 30 data tuples, with the remaining data used as the test set. For Qwen3-32B, due to its significantly higher inference cost, we use a smaller validation set of 20 data tuples. The hyperparameter $\alpha$ is selected from $\{1, 5, 10, 15, 20, 30, 40\}$ based on the Sliced Shapley value computed on the corresponding validation set. We do not compare with DuoAttention (Xiao et al., 2025a) because its requirement for certain attention heads to retain the full key-value cache exceeds the total budget constraint of our eviction policy.

### 4.2 HYPERPARAMETER FREE RESULTS.

Since both HeadKV-R2 and CoKV provide importance scores for each group, we conduct an experiment to compare their effectiveness without introducing any additional hyperparameters. In this experiment, we mask the caches of groups based on the importance scores assigned by each algorithm. This experiment can be viewed as a specific case of adaptive KV quantization that switches between 0 bits and 16 bits. Specifically, we mask the caches of both the highest-ranked (**top**) and lowest-ranked groups (**low**). The complete results are shown in Tables 16, 17 and 18 in the appendix. We include a

Table 1: Comprehensive Masking Top Important Heads Results on LongBench

| Method | Single-Doc. QA | | | Multi-Doc. QA | | | Summarization | | | Few-shot Learning | | | Synthetic | | Code | | Avg |
|---|---|---|---|---|---|---|---|---|---|---|---|---|---|---|---|---|---|
| | NtrQA | Qasper | MF-en | HotpotQA | 2WikiMQA | Musique | GovReport | QMSum | MultiNews | TREC | TriviaQA | SAMSum | PCount | PRe | Lcc | RB-P | |
| **Masking Top 16 heads with Llama-3-8B-Instruct model** | | | | | | | | | | | | | | | | | |
| Full Cache | 24.12 | 31.24 | 39.85 | 45.23 | 34.56 | 21.09 | 28.38 | 23.24 | 26.52 | 74.12 | 90.96 | 42.37 | 4.55 | 71.76 | 58.10 | 51.64 | 41.73 |
| Random | 20.93 | 28.48 | 33.69 | 44.93 | 20.01 | 20.6 | 28.43 | 23.7 | 26.67 | 74.12 | 91.07 | 41.12 | 4.26 | 71.76 | 49.83 | 40.55 | 38.76 |
| HeadKV-R2(top) | 19.45 | 12.97 | 27.75 | 34.2 | 17.33 | 14.32 | 19.74 | 22.76 | 22.05 | 67.06 | 87.91 | 35.53 | 4.71 | 68.49 | 26.62 | 26.53 | 31.71 |
| CoKV(top) | 6.55 | 9.46 | 9.47 | 10.19 | 12.27 | 5.67 | 5.73 | 16.96 | 4.47 | 43.53 | 71.21 | 23.77 | 3.91 | 34.98 | 11.58 | 17.18 | 17.93 |
| **Masking Top 64 heads with Qwen3-32B model** | | | | | | | | | | | | | | | | | |
| Full Cache | 37.14 | 45.51 | 49.17 | 58.2 | 54.74 | 38.36 | 32.9 | 23.72 | 25.08 | 72.78 | 72.57 | 37.95 | 17.78 | 100 | 62.72 | 70.08 | 49.92 |
| Random | 30.13 | 44.81 | 50.07 | 56.51 | 54.06 | 39.24 | 27.33 | 23.32 | 24.82 | 72.78 | 71.12 | 37.72 | 12.94 | 100.0 | 22.71 | 23.79 | 43.21 |
| HeadKV-R2(top) | 28.38 | 34.55 | 32.15 | 47.25 | 45.26 | 25.33 | 20.14 | 22.17 | 13.98 | 55.56 | 56.69 | 21.18 | 8.69 | 98.33 | 14.44 | 18.47 | 33.91 |
| CoKV(top) | 28.85 | 28.55 | 18.91 | 25.1 | 19.78 | 12.69 | 13.48 | 22.97 | 23.8 | 35.56 | 26.43 | 10.41 | 8.04 | 17.22 | 5.2 | 5.05 | 18.88 |

simplified table for the results of masking groups of Qwen3-32B and Llama-3-8B-Instruct model in Table 1. The results show that when masking the top-ranked groups identified by each method, the performance of CoKV degrades more significantly than that of HeadKV-R2. This suggests that CoKV is more effective at ranking group importance, as it better distinguishes between critical and non-critical caches. Conversely, the results in the full tables show that when masking the unimportant groups (**low**), the performance of CoKV declines more gradually than HeadKV-R2. When masking the 64 most important heads of Qwen3-32B, CoKV achieves an average accuracy of only 18.88%, while HeadKV maintains 33.91%. This demonstrates that CoKV more accurately identifies critical heads, as its performance drops more significantly when they are removed. Suprisingly, the results of masking 16 most unimportant groups in Table 16 and 18 outperformed the FullKV approach. This further demonstrates that CoKV can identify groups that have a negative impact on the model. By removing the KV pairs from these groups, the model inference not only optimizes storage and decoding speed but also enhances overall performance.

## 4.3 KV Cache Eviction Results

**Benchmark Results.** The complete benchmark results are presented in Tables 14 and 15 in the appendix. We include a simplified table (Table 2), showing the performance of Llama-3-8B-Instruct and Qwen3-32B when keeping 64 KV pairs on average for Llama-3-8B-Instruct and 128 KV pairs on average for Qwen3-32B. The results demonstrate that CoKV consistently outperforms all baseline methods. The superior performance of CoKV arises from its ability to effectively evaluate the

Table 2: Comprehensive KV Cache Eviction Results on LongBench

| Method | Single-Doc. QA | | | Multi-Doc. QA | | | Summarization | | | Few-shot Learning | | | Synthetic | | Code | | Avg |
|---|---|---|---|---|---|---|---|---|---|---|---|---|---|---|---|---|---|
| | NtrQA | Qasper | MF-en | HotpotQA | 2WikiMQA | Musique | GovReport | QMSum | MultiNews | TREC | TriviaQA | SAMSum | PCount | PRe | Lcc | RB-P | |
| **Llama-3-8B-Instruct model with KV size=64** | | | | | | | | | | | | | | | | | |
| Full Cache | 24.12 | 31.24 | 39.85 | 45.23 | 34.56 | 21.09 | 28.38 | 23.24 | 26.52 | 74.12 | 90.96 | 42.37 | 4.55 | 71.76 | 58.1 | 51.64 | 41.73 |
| SnapKV | 19.94 | 13.21 | 28.91 | 40.06 | 28.58 | 18.12 | 17.29 | 21.71 | 17.05 | 49.41 | 89.00 | 35.48 | 3.99 | 71.57 | 54.35 | 50.42 | 34.94 |
| Pyramid | 20.11 | 16.54 | 32.67 | 40.25 | 27.71 | 17.54 | 18.67 | 22.37 | 20.03 | 62.55 | 89.89 | 36.63 | 4.30 | 71.76 | 54.27 | 50.96 | 36.64 |
| Ada-SnapKV | 20.40 | 14.46 | 32.62 | 42.39 | 31.48 | 17.58 | 18.57 | 22.18 | 18.71 | 58.82 | 90.13 | 35.25 | 4.41 | 71.57 | 54.02 | 51.68 | 36.52 |
| HeadKV-R2 | 20.30 | 16.76 | 35.96 | 38.08 | 26.41 | 17.98 | 18.68 | 21.75 | 20.58 | 67.06 | 88.19 | 37.30 | 3.21 | 71.76 | 56.20 | 54.49 | 37.17 |
| CoKV | 20.77 | 19.67 | 35.11 | 44.37 | 34.36 | 17.83 | 17.89 | 22.33 | 18.55 | 71.76 | 90.73 | 38.51 | 4.71 | 71.76 | 55.45 | 55.82 | 38.73 |
| **Qwen3-32B model with KV size=128** | | | | | | | | | | | | | | | | | |
| Full Cache | 37.14 | 45.51 | 49.17 | 58.2 | 54.74 | 38.36 | 32.9 | 23.72 | 25.08 | 72.78 | 72.57 | 37.95 | 17.78 | 100.0 | 62.72 | 70.08 | 49.92 |
| SnapKV | 30.18 | 32.54 | 41.96 | 55.2 | 47.04 | 34.31 | 22.74 | 21.19 | 19.0 | 47.22 | 67.89 | 36.77 | 17.22 | 98.89 | 58.52 | 50.48 | 42.65 |
| Pyramid | 27.17 | 32.11 | 40.93 | 30.13 | 37.2 | 34.43 | 22.01 | 20.95 | 18.71 | 38.1 | 68.43 | 35.55 | 9.52 | 100.0 | 56.48 | 51.29 | 40.19 |
| Ada-SnapKV | 22.74 | 32.11 | 40.02 | 32.42 | 40.44 | 27.97 | 23.43 | 21.67 | 18.98 | 46.67 | 68.43 | 38.93 | 9.52 | 100.0 | 57.9 | 49.71 | 39.43 |
| HeadKV-R2 | 29.78 | 33.39 | 41.51 | 51.85 | 52.06 | 36.17 | 24.17 | 21.04 | 19.03 | 48.89 | 67.98 | 36.44 | 18.02 | 99.44 | 56.88 | 52.47 | 43.07 |
| CoKV | 28.73 | 37.56 | 44.80 | 53.48 | 53.07 | 35.17 | 23.89 | 21.34 | 19.28 | 54.44 | 69.35 | 38.23 | 19.22 | 100.0 | 57.13 | 56.87 | 44.53 |

importance of each cache within a group while considering the cooperation among all groups. It is not only capable of identifying which groups are important but also able to recognize those groups that do not contribute or even have a negative contribution. By optimizing the cache size to enhance overall cooperation, CoKV ensures efficient and high-quality inference.

**Decoding Latency and Memory Usage**   We conduct experiments using the Qwen3-32B model, which supports a maximum context window of 128k tokens with YaRN (Peng et al.), with FlashAttention enabled as the default setting. All methods except FullKV, were executed on a single H100 96G GPU. The FullKV method alone necessitated the use of two H100 96G GPUs. We design two key experiments with the average KV cache size set to 128 tokens for all KV cache eviction methods.

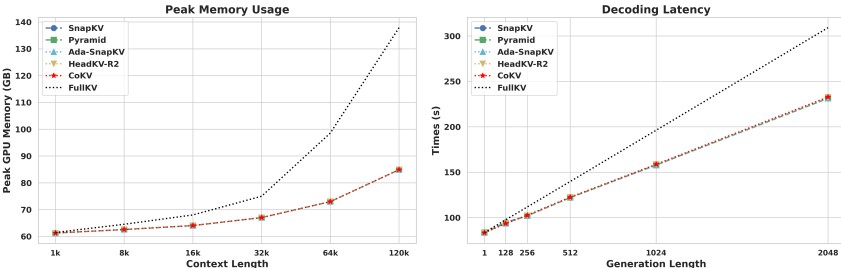

Figure 2: Results of Peak Memory Usage and Decoding Latency.

**Peak Memory Usage**   Under fixed generation length (1 token), we measure the peak GPU memory usage (including model parameters and runtime states) across varying input contexts (1k/8k/16k/32k/64k/120k tokens). As shown in the Peak Memory Usage of Figure 2, CoKV reduces memory usage by 38.4% compared to FullKV baseline at 120k input length. Notably, CoKV can accommodate 120k inputs on a single H100 GPU, in contrast to FullKV, which supports under 60k.

**Decoding Latency**   With a fixed input context length of 120k tokens, we measure decoding latency (including both the pre-filling time and the decoding time) across different generation lengths (1/128/256/512/1024/2048 tokens). As shown in the Decoding Latency of Figure 2, CoKV achieves less than 25.14% of the total latency compared to the FullKV baseline, with negligible differences observed between the other baselines(comparative experiments showed less than 2% variation across 64/256/512/1024 tokens).

## 4.4 KV CACHE QUANTIZATION RESULTS

To evaluate the effectiveness of CoKV, we conducted comparative experiments under a KV cache quantization setting. Specifically, we compared CoKV against HeadKV-R2 using an adaptive quantization strategy with two bit-widths. The more important half of the attention heads were quantized to 8 bits, and the remaining half to 4 bits. The results are presented in Table 3. We also conduct KV quantization with Qwen model which is shown in Table 19 in the appendix.

Table 3: Llama-3-8B Instruct Model with 8-4 Bits Quantization

| Method | Single-Doc. QA | | | Multi-Doc. QA | | | Summarization | | | Few-shot Learning | | | Synthetic | | Code | | Avg |
|---|---|---|---|---|---|---|---|---|---|---|---|---|---|---|---|---|---|
| | NtrQA | Qasper | MF-en | HotpotQA | 2WikiMQA | Musique | GovReport | QMSum | MultiNews | TREC | TriviaQA | SAMSum | PCount | PRe | Lcc | RB-P | |
| HeadKV-R2 | 21.23 | 31.86 | 35.33 | 43.51 | 34.44 | 17.05 | 17.19 | 15.44 | 15.23 | 73.53 | 74.6 | 31.83 | 7.14 | 35.71 | 58.86 | 43.29 | 34.77 |
| CoKV | 22.58 | 32.62 | 38.36 | 44.91 | 34.53 | 21.61 | 16.9 | 17.35 | 15.68 | 75.29 | 82.14 | 31.02 | 7.14 | 35.71 | 60.29 | 42.64 | 36.17 |

## 5 CONCLUSION

Large language models (LLMs) face significant challenges in inference cost due to the excessive memory and latency overhead associated with the growing size of the KV cache. To this end, we introduce CoKV, a novel method designed to evaluate the collaborative importance of attention heads and dynamically allocate cache sizes based on Sliced Shapley value. Our experimental results demonstrate that CoKV achieves state-of-the-art performance across 16 LongBench datasets, as well as on the NIAH, RULER, and math reasoning benchmarks. CoKV provides a scalable and practical solution for enhancing the efficiency of LLMs in real-world applications.

ETHICS STATEMENT

This study uses only public benchmark datasets and open-source models. No personal data or sensitive attributes are involved, and use follows the original dataset terms. The research does not involve any personal, private, or sensitive data. The proposed methods are intended for research purposes, and we have conducted a review which identified no foreseeable ethical concerns.

REPRODUCIBILITY STATEMENT

All datasets and models used in this study are publicly accessible online. The experimental details are described in the Experiment section, and the code for reproducing all results is available in our anonymous repository.

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

APPENDIX

## A  USE OF LARGE LANGUAGE MODELS

In accordance with the ICLR policy on large language model usage, we report the use of LLMs in the preparation of this manuscript. We used GPT-5 as a writing assistance tool to improve grammar and clarity. We also used Cursor for code completion and debugging. The authors reviewed and tested all code. LLMs did not contribute to the design of any methods or the implementation of core algorithms. The LLM was not used for research ideation or the design of methods. All scientific ideas, experimental designs, theoretical results, and conclusions presented in this paper are entirely the work of the authors.

## B  RELATED WORKS

**KV Cache Compression**  The memory overhead of storing key-value (KV) pairs for LLM has motivated extensive research on KV cache compression. StreamingLLM Xiao et al. (2024) preserves the initial and recent tokens, which empirically exhibit higher informativeness during generation. Similarly, Scissorhands Liu et al. (2023) proposes the persistence of importance to identify and retain pivotal tokens. H2O Zhang et al. (2023b) employs a heavy-hitter oracle to drop tokens with low attention scores.SnapKV Li et al. (2024b) uses the attention scores of the recent tokens to retain critical tokens. CCA-LLM Chen et al. (2024c) groups input tokens, compressing each group into a core token, which are then combined with recent tokens for attention computation. DGA-LLM Zhang et al. (2025) aggregates less important tokens while preserving important tokens. While these methods reduce memory usage and accelerate inference, they implicitly assume uniform importance across attention heads, limiting their applicability. Recent works address head diversity through layer-wise and head-wise optimizations. PyramidKV Cai et al. (2024) implements a hierarchical allocation strategy, assigning larger cache budgets to lower layers based on the observed attention patterns across layers. FastGen Ge et al. (2024) is an adaptive KV cache compression method that reduces LLMs' memory usage by profiling attention modules and constructing caches adaptively. RazorAttention Tang et al. (2025) and Duoattention Xiao et al. (2025a) divide attention heads into retrieval heads(critical for long-context processing Wu et al. (2025)) and non-retrieval heads, apply full KV cache to retrieval heads and compressed KV cache to non-retrieval heads. ArkVale Chen et al. (2024b) proposes a page-based KV cache manager that asynchronously copies filled pages into external memory (e.g., CPU memory) as a backup and supports the recall of important tokens that were previously evicted. AdaKV Feng et al. (2025) dynamically adjusts cache budgets across heads based on their concentration degrees and HeadKV Fu et al. (2025) calculates head importance scores to allocate cache budget before inference. However, these methods assess heads in isolation, neglecting their collaborative interactions. For example, the standalone score of a head may not reflect its true contribution when working synergistically with others. Additionally, these approaches overlook the task-dependent variations in head importance. Our approach tackles these limitations by modeling head interactions as a cooperative game, dynamically allocating cache resources based on the varying complementary contributions of heads across different tasks.

In addition to KV cache eviction methods, KV cache quantization is also one of the mainstream approaches for KV cache compression Yang et al. (2024); Liu et al. (2024). However, while eviction methods can be used to retain less than 1% of the cache, KV cache compression cannot be applied to such an extent because it must preserve at least 1 bit. Nevertheless, the combination of these two methods is an interesting direction for future research.

**Model Architecture and Computation Optimization**  Modern LLMs employ architectural optimizations to balance efficiency and performance. Multi Query Attention (MQA) Shazeer (2019) shares a single key-value pair across all attention heads, drastically reducing memory usage but potentially sacrificing performance. Group Query Attention (GQA) Ainslie et al. (2023) strikes a balance by grouping heads to share key-value pairs, preserving performance while maintaining memory efficiency, which is widely adopted in recent LLMs like Llama Dubey et al. (2024) and Mistral Jiang et al. (2023). Concurrently, Flash Attention Dao et al. (2022) optimizes hardware utilization by minimizing memory reads/writes during attention computation, significantly accelerat-

ing inference. Notably, our approach is fully compatible with GQA and Flash Attention, ensuring seamless integration with current LLMs.

**Cooperative Game Theory** Cooperative game theory offers a principled framework for understanding how multiple players can jointly contribute to overall system performance. Shapley value Shapley (1953), a classic solution in cooperative game theory, provides a method for fairly allocating joint benefits based on the marginal contribution of each player. However, traditional Shapley value computation methods allow each sample to be used to calculate the marginal contribution of only a single player. Recent works Zhang et al. (2023a); Sun et al. (2024) address this limitation through complementary contributions that enable simultaneous estimation of multiple players' contributions. In the context of LLMs, these methods still encounter scalability issues, as the cost of computing complementary contributions across all coalition sizes remains prohibitively high. We propose the Sliced Shapley value, which samples only a subset of coalition sizes. This approach not only accelerates the computation but also accurately reflects the contributions of different heads.

## C SUPLEMENTARY OF PRELIMINARIES

### C.1 KV CACHE EVICTION

KV cache eviction methods can be employed to discard unimportant KV cache entries while preserving model performance. As the attention heads process more tokens, the KV cache can grow in size, which results in increased memory usage and computation costs. To address this, selective eviction methods can be introduced to remove KV pairs that contribute less to the final attention results. Typically, the eviction is based on criteria such as the relevance of key-value pairs (e.g., low activation values or relevance scores) or certain pruning strategies based on model performance.

For each head $h_i$, the compressed KV cache is reduced to $\hat{K}_i \in \mathbb{R}^{s \times d_h}$ and $\hat{V}_i \in \mathbb{R}^{s \times d_h}$, where some unimportant KV pairs are evicted, and $s \ll m$, resulting in a significant improvement in computational efficiency and memory usage. This compression is typically done by selecting the most relevant KV pairs and discarding the rest. The process is often repeated over multiple layers or tokens, progressively reducing the size of the KV cache while maintaining performance.

Specifically, the compressed KV cache is updated by appending the new key and value pair:

$$\hat{K}_i = \text{Cat}[\hat{K}_i, k_i], \quad \hat{V}_i = \text{Cat}[\hat{V}_i, v_i].$$

The attention output for each head $h_i$ is computed using the compressed KV cache by $\hat{O}_i = \hat{A}_i \hat{V}_i$, where the attention weights $\hat{A}_i$ are calculated as:

$$\hat{A}_i = \text{softmax}(q_i \hat{K}_i^T / \sqrt{d_h}).$$

By selectively discarding less relevant KV pairs, the model can maintain a more efficient cache, reducing memory and computation overhead. The effectiveness of this method depends on how well the eviction process retains the most important KV pairs for accurate attention calculation, ensuring that the overall model performance remains optimal despite the reduced cache size.

### C.2 KV CACHE QUANTIZATION

The uniform quantization to the KV cache process is shown as follows. For each head group $h_i$ with bit-width $b_i$, and for each token position $t$, we compute a per-token dynamic range along the head dimension. Define the integer range as:

$$q_{\max}(b_g) = \max\left(1, 2^{b_i - 1} - 1\right), \quad b_i > 1$$

and the scale as:

$$s_{i,t} = \frac{\max|K_{i,t}|}{q_{\max}(b_i)}, \quad u_{i,t} = \frac{\max|V_{i,t}|}{q_{\max}(b_i)}$$

Uniform quantization maps float element $K_{i,t,j}(1 \leq j \leq d_h)$ to integers via:

$$\overline{K}_{i,t,j} = \text{clip}\left(\text{round}\left(\frac{K_{i,t,j}}{s_{i,t}}\right), -q_{\max}(b_i), q_{\max}(b_i)\right)$$

and for values:

$$\overline{V}_{i,t,j} = \text{clip}\left(\text{round}\left(\frac{V_{i,t,j}}{u_{i,t}}\right), -q_{\max}(b_i), q_{\max}(b_i)\right)$$

The dequantization process restores the values by:

$$\tilde{K}_{i,t,j} = \overline{K}_{i,t,j} \times s_{i,t}, \quad \tilde{V}_{i,t,j} = \overline{V}_{i,t,j} \times u_{i,t}$$

Specifically, the compressed KV cache is updated by appending the new key and value pair:

$$\tilde{K}_i = \text{Cat}[\tilde{K}_i, k_i], \quad \tilde{V}_i = \text{Cat}[\tilde{V}_i, v_i].$$

The attention output for each head $h_i$ is computed using the compressed KV cache by $\tilde{O}_i = \tilde{A}_i \tilde{V}_i$, where the attention weights $\tilde{A}_i$ are calculated as:

$$\tilde{A}_i = \text{softmax}(q_i \tilde{K}_i^T / \sqrt{d_h}).$$

Since we only cache the quantized matrices $\overline{K}_i$ and $\overline{V}_g$ which use $b_i$ bits for storing each element, it can significantly reduce memory usage during model inference.

## D  PROOF

### D.1  PROOF OF THEOREM 1

Let $\mathcal{N} = \{1, 2, \ldots, n\}$ be the set of coalition sizes, and let $Z_j = \mathcal{SV}_{\{i,j\}}$ for $j \in \mathcal{N}$, where $\mathcal{SV}_{i,j}$ denotes the expected complementary contribution for coalition size $j$. Assume $Z_j \in [a, b]$ for all $j$, and define $R = b - a$. The true Shapley value is the population mean:

$$\mu = \frac{1}{n} \sum_{j=1}^{n} Z_j = \mathcal{SV}_i.$$

For a randomly sampled subset $\mathcal{H} \subseteq \mathcal{N}$ of size $|\mathcal{H}|$ drawn without replacement, the Sliced Shapley value is the sample mean:

$$\bar{Z} = \frac{1}{|\mathcal{H}|} \sum_{j \in \mathcal{H}} Z_j = \mathcal{SSV}_i^{\mathcal{H}}.$$

To bound the error $|\bar{Z} - \mu|$, we apply Hoeffding's inequality for sampling without replacement. Consider a random permutation $\pi$ of $\mathcal{N}$, and let $X_i = Z_{\pi(i)}$ for $i = 1, \ldots, |\mathcal{H}|$. The sample sum is $S = \sum_{i=1}^{|\mathcal{H}|} X_i$. By Serfling's result (Serfling, 1974), the moment generating function satisfies:

$$E[\exp(\lambda(S - \mu))] \leq \exp\left(\frac{\lambda^2 |\mathcal{H}| R^2}{8} \frac{n}{n - |\mathcal{H}| + 1}\right).$$

Applying the Chernoff bound (Chernoff, 1952), for any $t > 0$:

$$P(S - \mu \geq t) \leq \min_{\lambda > 0} E[\exp(\lambda(S - \mu))] \exp(-\lambda t).$$

Substituting the Serfling's bound:

$$P(S - \mu \geq t) \leq \min_{\lambda > 0} \exp\left(\frac{\lambda^2 |\mathcal{H}| R^2}{8} \frac{n}{n - |\mathcal{H}| + 1} - \lambda t\right).$$

The expression on the right is minimized by choosing $\lambda = \frac{4t}{|\mathcal{H}|R^2} \frac{n-|\mathcal{H}|+1}{n}$. Plugging this value in yields:

$$P(S - \mu \geq t) \leq \exp\left(-\frac{2t^2}{|\mathcal{H}|R^2} \frac{n}{n-|\mathcal{H}|+1}\right).$$

By symmetry,

$$P(|S - |\mathcal{H}|\mu| \geq t) \leq 2\exp\left(-\frac{2t^2}{|\mathcal{H}|R^2} \frac{n}{n-|\mathcal{H}|+1}\right).$$

Substituting $t = |\mathcal{H}|\epsilon$ yields:

$$P(|S - |\mathcal{H}|\mu| \geq |\mathcal{H}|\epsilon) = P(|S - |\mathcal{H}|\mu| \geq |\mathcal{H}|\epsilon) \leq 2\exp\left(-\frac{|\mathcal{H}|\epsilon^2}{R^2} \frac{n}{n-|\mathcal{H}|+1}\right).$$

Setting the right-hand side to $\delta$ and solving for $\epsilon$:

$$2\exp\left(-\frac{2|\mathcal{H}|\epsilon^2}{R^2} \frac{n}{n-|\mathcal{H}|+1}\right) = \delta,$$

$$\ln(2) - \frac{2|\mathcal{H}|\epsilon^2}{R^2} \frac{n}{n-|\mathcal{H}|+1} = \ln(\delta),$$

$$\epsilon^2 = \frac{R^2}{2|\mathcal{H}|}(n - |\mathcal{H}| + 1)\ln\left(\frac{2}{\delta}\right).$$

Thus, with probability at least $1 - \delta$:

$$|\mathcal{SV}_i - \mathcal{SSV}_i^{\mathcal{H}}| \leq R\sqrt{\frac{(n - |\mathcal{H}| + 1)\ln(2/\delta)}{2|\mathcal{H}|n}}.$$

It is clear that $\frac{(n-\mathcal{H}+1)}{n} \leq 1$ as $\mathcal{H} \geq 1$. Therefore, we have

$$|\mathcal{SV}_i - \mathcal{SSV}_i^{\mathcal{H}}| \leq R\sqrt{\frac{\ln(2/\delta)}{2|\mathcal{H}|}}.$$

By omitting the constants, we obtain the asymptotic error bound $\mathcal{O}(|\mathcal{SV}_i - \mathcal{SSV}_i^{\mathcal{H}}|)$. This completes the proof.

### D.2 PROOF OF THEOREM 2

In this section, we give the proof of Theorem 2. Denote $\mathcal{H}$ the selected coalition sizes. The approximation of $\mathcal{SV}_{i,j}(1 \leq i, j \leq n)$ is unbiased, which can be proven following Corollary 1 in Sun et al. (2024). So it is evident that $\overline{\mathcal{SSV}_i}$, being the weighted average of $\overline{\mathcal{SV}_{i,j}}$, serves as an unbiased estimator of $\mathcal{SSV}_i$. Hence, we have

$$\mathbb{P}(|\overline{\mathcal{SSV}_i^{\mathcal{H}}} - \mathcal{SSV}_i^{\mathcal{H}}| \geq \epsilon) \leq \mathbb{P}(\sum_{j \in \mathcal{H}} |\overline{\mathcal{SV}_{i,j}} - \mathcal{SV}_{i,j}| \geq \epsilon)$$

$$\leq \sum_{j \in \mathcal{H}} \mathbb{P}(|\overline{\mathcal{SV}_{i,j}} - \mathcal{SV}_{i,j}| \geq \frac{\epsilon}{|\mathcal{H}|})$$

Then,we have

$$\sum_{j \in \mathcal{H}} \mathbb{P}(|\overline{\mathcal{SV}_{i,j}} - \mathcal{SV}_{i,j}| \geq \frac{\epsilon}{|\mathcal{H}|}) \leq 2|\mathcal{H}| \exp(-\frac{2(\frac{\epsilon}{|\mathcal{H}|})^2}{\sum_{k=1}^{\mathcal{M}/|\mathcal{H}|}(b_j - a_j)^2})$$

$$\leq 2|\mathcal{H}| \exp(-\frac{2(\frac{\epsilon}{\mathcal{H}})^2}{\frac{\mathcal{M}r^2}{|\mathcal{H}|}}),$$

according to Hoeffding's inequality where $(a_j, b_j)$ denotes the range of complementary contribution of j-coalitions, and r is $\max(b_1 - a_1, \cdots, b_j - a_j)$. . Since we want the right hand side to be at most $\delta$, we have $\mathcal{M} \geq \frac{\mathcal{H}r^2 ln \frac{2\mathcal{H}}{\delta}}{2\epsilon^2}$. Thus, Alogorithm 1 returns an $(\epsilon, \delta)$-approximation of Sliced Shapley value with time complexity $\mathcal{O}(\frac{T|\mathcal{H}|ln \frac{2|\mathcal{H}|}{\delta}}{\epsilon^2})$ where T is the time cost of evaluating each complementary contribution. The analysis of the time complexity of approximating Shapley value starts from $\mathbb{P}(|\overline{\mathcal{SV}_1} - \mathcal{SV}_i| \geq \epsilon) \leq \mathbb{P}(\sum_{j=1}^n |\overline{\mathcal{SV}_{i,j}} - \mathcal{SV}_{i,j}| \geq \epsilon)$ Following similar steps, we can proof that the time complexity of approximating Shapley value is $\mathcal{O}(\frac{Tnln \frac{2n}{\delta}}{\epsilon^2})$. Thus, we complete the proof.

# E SUPPLEMENTARY EXPERIMENTS

## E.1 DATASETS

**LongBench** (Bai et al., 2023) is a multitask benchmark for long context understanding and exhibits a wide range of average input lengths, spanning from 1,235 to 18,409 tokens. We introduce the detailed information of LongBench in Table 4, including the task types, evaluation metrics, average length, languages, and the number of samples for each task. .

Table 4: Details of 16 Datasets in LongBench

| Label | Task Type | Eval metric | Avg len | Language | Sample Num |
|---|---|---|---|---|---|
| NrtvQA | Single-Doc. QA | F1 | 18,409 | EN | 200 |
| Qasper | Single-Doc. QA | F1 | 3,619 | EN | 200 |
| MF-en | Single-Doc. QA | F1 | 4,559 | EN | 150 |
| HotpotQA | Multi-Doc. QA | F1 | 9,151 | EN | 200 |
| 2WikiMQA | Multi-Doc. QA | F1 | 4,887 | EN | 200 |
| Musique | Multi-Doc. QA | F1 | 11,214 | EN | 200 |
| GovReport | Summarization | Rouge-L | 8,734 | EN | 200 |
| QMSum | Summarization | Rouge-L | 10,614 | EN | 200 |
| MultiNews | Summarization | Rouge-L | 2,113 | EN | 200 |
| TREC | Few-shot Learning | Accuracy | 5,177 | EN | 200 |
| TriviaQA | Few-shot Learning | F1 | 8,209 | EN | 200 |
| SAMSum | Few-shot Learning | Rouge-L | 6,258 | EN | 200 |
| PCount | Synthetic | Accuracy | 11,141 | EN | 200 |
| PRe | Synthetic | Accuracy | 9,289 | EN | 200 |
| Lcc | Code | Edit Sim | 1,235 | Python/C#/Java | 500 |
| RB-P | Code | Edit Sim | 4,206 | Python/Java | 500 |

**NIAH** (Needle In A Haystack) is a test specifically designed to evaluate a model's ability to locate and recall a small piece of critical information (the "needle") hidden within a very long, irrelevant text (the "haystack"). It systematically measures retrieval accuracy as the document length increases, directly probing the limits of a model's context window.
**RULER** is a comprehensive benchmark for evaluating the factual reasoning capabilities of large language models over long contexts. It expands tasks from the popular "LAMA" probe into a

long-context setting, challenging models to maintain accuracy when factual knowledge is distributed across thousands of tokens.

**GSM8K** (Cobbe et al., 2021) is a dataset of diverse grade-school level math word problems. Each problem requires a sequence of logical reasoning steps to solve, and the benchmark is primarily used to evaluate a model's fundamental mathematical reasoning and arithmetic capabilities.

**MATH** (Hendrycks et al., 2021) is a large-scale dataset containing challenging high-school competition-level mathematics problems. It tests advanced reasoning by requiring solutions to problems from various sub-fields like algebra, geometry, and calculus, often demanding step-by-step derivations.

### E.2 ABLATION STUDY

In this section, we analyze the number of coalition sizes in $\mathcal{H}$ using the Llama-3-8B Instruct model. We conducted experiments by masking the 32 most important attention heads (top) and the 32 least important heads (low) based on their significance scores. This approach eliminates the need for hyperparameter tuning, ensuring a fair comparison between CoKV and HeadKV by removing potential fluctuations caused by parameter selection. We compare the results for $\mathcal{H} = \{32, 64, 96, 128\}$ with $\mathcal{H} = \{128\}$. The results show that using the expected complementary contribution of a single coalition size 128 in Llama-3-8B-Instruct still outperforms the baselines, and CoKV is more effective with a larger number of coalition sizes. The results are shown in Table 5.

Table 5: Ablation Study on Llama-3-8B-Instruct models using LongBench

| Method | Single-Doc. QA | | | Multi-Doc. QA | | | Summarization | | | Few-shot Learning | | | Synthetic | | Code | | Avg |
|---|---|---|---|---|---|---|---|---|---|---|---|---|---|---|---|---|---|
| | NrtvQA | Qasper | MF-en | HotpotQA | 2WikiMQA | Musique | GovReport | QMSum | MultiNews | TREC | TriviaQA | SAMSum | PCount | PRe | Lcc | RB-P | |
| Full Cache | 24.12 | 31.24 | 39.85 | 45.23 | 34.56 | 21.09 | 28.38 | 23.24 | 26.52 | 74.12 | 90.96 | 42.37 | 4.55 | 71.76 | 58.10 | 51.64 | 41.73 |
| Random | 20.69 | 18.60 | 29.63 | 39.12 | 18.50 | 6.94 | 22.40 | 22.33 | 26.45 | 74.12 | 89.82 | 33.80 | 4.71 | 61.12 | 30.78 | 40.71 | 33.73 |
| HeadKV-R2(top) | 17.33 | 6.98 | 9.37 | 13.50 | 9.37 | 5.11 | 13.18 | 20.86 | 15.24 | 45.88 | 75.30 | 27.21 | 4.76 | 66.21 | 11.24 | 13.64 | 22.20 |
| CoKV(top) | 1.40 | 3.49 | 3.78 | 7.94 | 9.32 | 2.32 | 2.64 | 11.74 | 0.58 | 34.71 | 21.37 | 6.96 | 4.14 | 16.93 | 3.54 | 5.17 | 8.50 |
| CoKV(coalition 128)(top) | 5.98 | 4.41 | 3.98 | 9.25 | 13.02 | 2.80 | 5.16 | 10.92 | 2.23 | 33.58 | 24.15 | 8.87 | 7.36 | 15.89 | 3.67 | 8.94 | 10.01 |
| HeadKV-R2(low) | 21.51 | 11.16 | 25.33 | 19.52 | 14.48 | 7.42 | 16.73 | 23.91 | 14.58 | 74.12 | 89.09 | 40.69 | 4.66 | 70.09 | 33.13 | 32.39 | 31.18 |
| CoKV(low) | 22.45 | 33.06 | 38.34 | 45.82 | 39.62 | 20.18 | 28.39 | 24.04 | 26.67 | 74.12 | 91.14 | 41.70 | 4.71 | 71.76 | 52.24 | 64.94 | 42.45 |
| CoKV(coalition 128)(low) | 20.85 | 29.56 | 33.62 | 46.01 | 35.27 | 18.80 | 27.93 | 22.18 | 23.57 | 74.12 | 91.08 | 40.25 | 4.56 | 71.76 | 48.66 | 50.54 | 39.92 |

### E.3 COMPUTATION EFFICIENCY

We further conduct experiments to evaluate the efficiency of approximating the Sliced Shapley value (referred to as CoKV in our method) using the qasper dataset with the Qwen3-32B model. A subset of 20 examples from the qasper dataset is used as the validation set for computing the Sliced Shapley value which is the same as other experiments. The experiment is performed on a single H100 96GB GPU, with a coalition size of 256, consistent with the settings in our other experiments.

Table 6: Overlap Ratio of Top 50% Important Heads

| Sampling number | 70 | 80 | 140 | 160 | 210 | 240 |
|---|---|---|---|---|---|---|
| Time (hours) | 4.97 | 5.71 | 9.64 | 10.91 | 14.32 | 17.08 |
| Overlap Ratio (%) | 64.45 | 68.75 | 74.60 | 77.73 | 84.37 | 90.18 |

To assess the stability of the approximation, we introduce a new metric, the overlap ratio of the top 50% important attention heads between two independent sampling runs. As shown in Table 6, when the number of samples reaches 80, the overlap ratio exceeds two-thirds (68.75%), indicating that the CoKV importance scores are beginning to converge and can reliably identify the most influential heads. We recommend performing two independent sampling runs when computing CoKV. Once the overlap ratio of the top 50% important heads between the two runs exceeds two-thirds, the results can be averaged and used as the final importance scores. At this point, CoKV can effectively reflect the relative contributions of attention heads.

Notably, even for a large model like Qwen3-32B, the precomputation of CoKV importance requires only a few hours, a cost that is negligible compared to that of full training or fine-tuning. Once

computed, the importance scores can be stored and reused for long-term inference optimization, and shared across users of the same model architecture, making CoKV a highly cost-effective approach.

### E.4 DISTRIBUTION OF SLICED SHAPLEY VALUE

Figure 6 shows the distribution of Sliced Shapley values computed for the selected coalition size $\mathcal{H} = 256$ for Qwen3-32B on LongBench. Figures 7 and 8 illustrate the distribution for the selected coalition sizes $\mathcal{H} = 32, 64, 96, 128$ for Llama-3-8B-Instruct and Mistral-7B-v0.2, respectively.

We observe that the distributions of Sliced Shapley values exhibit significant differences across datasets of different task categories, while showing relatively smaller variations within datasets of the same domain type. In Figures 9 and 10, we present the distributions of the expected complementary contributions of heads in Llama-3-8B-Instruct model on the *hotpotqa* dataset (multi-document question answering) and the *lcc* dataset (code generation), with coalition sizes of $\{32, 64, 96, 128, 160, 192, 224\}$. We observe strong correlations in the distributions across all coalition sizes. Additionally, the distributions of the expected complementary contributions for coalition sizes $S$ and $n - |S|$ are nearly identical, exhibiting symmetry around the size of 128. To optimize computational efficiency, we restrict the calculation of complementary contributions to coalitions with sizes below 128. These observations provide a justification for our approach of computing complementary contributions using only a small subset of coalition sizes, as it effectively captures the contributions of the heads.

As showing all the distribution of all datasets costs too much pages, we conduct additional experiment analysis comparing the averaged complementary contributions of small coalitions (j=32,64,96) and large coalitions (j=160,192,224) by measuring the overlap rate of top-contributing heads between them in 16 datasets in LongBench. Specifically, we computed the percentage of shared heads in their respective top-k lists (k=32,64,128). The results show consistently high overlap rates (averaging over 85%) across all 16 datasets and two different LLMs, confirming that the distributional similarity is not model- or dataset-specific. This pattern suggests an underlying structural property of attention heads in transformer-based LLMs, where the complementary contribution of heads remains stable across different coalition scales. The results are shown in Tables 7 and 8.

Table 7: Overlap Results of Llama-3-8B-Instruct

| Method | Single-Doc. QA | | | Multi-Doc. QA | | | Summarization | | | Few-shot Learning | | | Synthetic | | Code | | Avg |
|---|---|---|---|---|---|---|---|---|---|---|---|---|---|---|---|---|---|
| | NrtQA | Qasper | MF-en | HotpotQA | 2WikiMQA | Musique | GovReport | QMSum | MultiNews | TREC | TriviaQA | SAMSum | PCount | PRe | Lcc | RB-P | |
| Top 32 heads | 0.81 | 0.66 | 0.81 | 0.91 | 0.84 | 0.81 | 0.62 | 0.66 | 0.50 | 0.84 | 0.84 | 0.59 | 0.97 | 0.97 | 0.81 | 0.75 | 0.77 |
| Top 64 heads | 0.77 | 0.81 | 0.88 | 0.88 | 0.91 | 0.91 | 0.67 | 0.72 | 0.66 | 0.88 | 0.77 | 0.80 | 0.98 | 0.98 | 0.83 | 0.86 | 0.83 |
| Top 128 heads | 0.82 | 0.88 | 0.80 | 0.93 | 0.90 | 0.95 | 0.83 | 0.83 | 0.80 | 0.95 | 0.84 | 0.80 | 0.99 | 0.99 | 0.88 | 0.85 | 0.88 |

Table 8: Overlap Results of Mistral-7B-Instruct-v0.2

| Method | Single-Doc. QA | | | Multi-Doc. QA | | | Summarization | | | Few-shot Learning | | | Synthetic | | Code | | Avg |
|---|---|---|---|---|---|---|---|---|---|---|---|---|---|---|---|---|---|
| | NrtQA | Qasper | MF-en | HotpotQA | 2WikiMQA | Musique | GovReport | QMSum | MultiNews | TREC | TriviaQA | SAMSum | PCount | PRe | Lcc | RB-P | |
| Top 32 heads | 0.66 | 0.84 | 0.72 | 0.78 | 0.75 | 0.94 | 0.75 | 0.72 | 0.62 | 0.75 | 0.78 | 0.72 | 1.00 | 0.94 | 0.75 | 0.69 | 0.78 |
| Top 64 heads | 0.77 | 0.94 | 0.77 | 0.72 | 0.81 | 0.94 | 0.92 | 0.72 | 0.69 | 0.86 | 0.84 | 0.77 | 0.97 | 0.94 | 0.72 | 0.73 | 0.82 |
| Top 128 heads | 0.83 | 0.94 | 0.84 | 0.86 | 0.87 | 0.97 | 0.95 | 0.84 | 0.88 | 0.88 | 0.87 | 0.86 | 0.99 | 0.95 | 0.86 | 0.80 | 0.89 |

### E.5 DECODING LATENCY AND MEMORY USAGE

We also conduct decoding latency and memory usage experiments using the Mistral-7B-Instruct-v0.2 model, which supports a maximum context window of 32k tokens, with FlashAttention enabled as the default setting, on an A100 GPU with 40GB of memory. We design two key experiments with the average KV cache size set to 128 tokens(comparative experiments showed less than 2% variation across 64/256/512/1024 tokens).

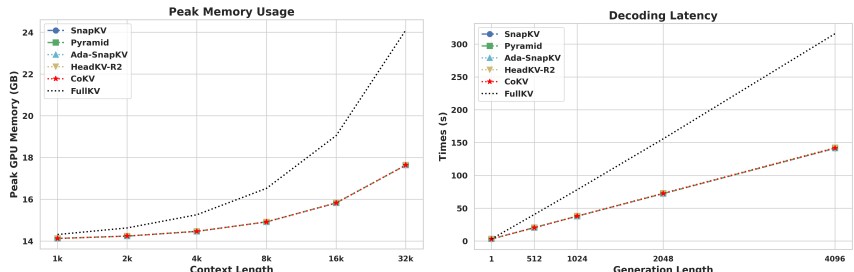

Figure 3: Results of Decoding Latency and Peak Memory Usage.

**Peak Memory Usage**   Under fixed generation length (1 token), we measure the peak GPU memory usage (including model parameters and runtime states) across varying input contexts (1k/2k/4k/8k/16k/32k tokens). As shown in the Peak Memory Usage of Figure 3, CoKV reduces memory usage by 64% compared to FullKV baseline at 32k input length.

**Decoding Latency**   With a fixed input context length of 28k tokens, we measure decoding latency (including both the pre-filling time and the decoding time) across different generation lengths (1/512/1024/2048/4096 tokens). As shown in the Decoding Latency of Figure 3, CoKV achieves less than 50% of the total latency compared to the FullKV baseline, with negligible differences observed between the other baselines.

### E.6   MATHEMATICAL REASONING EVALUATION

To assess the mathematical reasoning capability of CoKV, we evaluate it on the GSM8K and MATH datasets using the Qwen3-32B model. The head importance scores are derived from the average scores across all tasks in LongBench. We perform experiments involving both KV cache eviction and head masking. We use 5 shots in all the experiments. The results are shown in Tables 9, 10, 11,and 12.

Table 9: Qwen3-32B mode with KV size = 128 on GSM8K

| Full Cache | SnapKV | Pyramid | AdaKV | HeadKV-R2 | CoKV |
|---|---|---|---|---|---|
| 93.47 | 70.71 | 69.56 | 68.69 | 79.44 | 82.10 |

Table 10: Mask Top Important Heads Results on GSM8K

|  | 16 heads | 32 heads | 64 heads | 96 heads |
|---|---|---|---|---|
| Random | 92.24 | 91.73 | 89.57 | 88.15 |
| HeadKV-R2 | 91.24 | 90.38 | 72.15 | 36.85 |
| CoKV(top) | 86.57 | 19.40 | 3.77 | 1.89 |

### E.7   GENERALIZATION ANALYSIS

We first compute a general head importance score by averaging the scores across all tasks in Long-Bench using Qwen3-32B. This general score is then applied uniformly in head masking experiments for each individual task in LongBench. The results, summarized in Table 17, indicate that although CoKV using this general score performs better than using task-specific scores, it evaluates head importance across more tasks therefore leading to a more comprehensive assessment. Besides, the experiments in Section E.6 demonstrate that the head importance scores derived from LongBench generalize effectively to mathematical reasoning tasks, confirming that CoKV captures a robust and transferable notion of head importance.

Table 11: Qwen3-32B mode with KV size = 128 on MATH

| Full Cache | SnapKV | Pyramid | AdaKV | HeadKV-R2 | CoKV |
|---|---|---|---|---|---|
| 52.53 | 39.10 | 37.58 | 41.02 | 43.65 | 46.17 |

Table 12: Mask Top Important Heads Results on MATH

|  | 16 heads | 32 heads | 64 heads | 96 heads |
|---|---|---|---|---|
| Random | 52.15 | 51.06 | 46.97 | 42.39 |
| HeadKV-R2 | 51.41 | 36.54 | 32.83 | 10.61 |
| CoKV(top) | 45.45 | 21.37 | 2.92 | 1.70 |

To validate the generalization capability of our method, we also conduct cross-dataset evaluations on two task categories: 1. Multi-Document QA including 2WikiMQA and Musique datasets. 2. Code Processing including Lcc and RB-P datasets. Following Section 4.2, we mask top and low-ranked attention heads but cross-apply head importance scores between datasets within the same task (e.g., mask 2WikiMQA using Musique-derived scores). As shown in Table 20 and Table 21, our method maintains superior accuracy over baselines across both models, confirming that learned importance scores can generalize across datasets within shared task domains.

### E.8 NEEDLE-IN-A-HAYSTACK TEST

To evaluate the performance of different KV cache compression methods in long-context retrieval tasks, we conduct a Needle-in-a-Haystack benchmark test using the Mistral-7B-v0.2 model. With the average KV cache size 128, we systematically insert target texts (needles) at ten equidistant positions (11%, 22%, ..., 100%) across varying context lengths ranging from 1,000 to 31,000 tokens (in 1,000-token increments). As shown in Figure 11, CoKV outperforms other baseline methods, achieving an average score of 95.89% - the closest performance to the uncompressed FullKV benchmark.

We further extend our evaluation to the Qwen3-32B model under the same KV cache budget (average size = 128). We use the average head scores on LongBench as the key-value allocation score, which demonstrates that CoKV provides a fair measure of the retrieval ability of each head across different tasks. Needles are inserted at the same ten relative positions across context lengths ranging from 1,000 to 61,000 tokens (in 4,000-token increments). As illustrated in Figure 12, CoKV again demonstrates superior performance, attaining an average accuracy of 73.86%—the highest among all compressed methods and the closest to the FullKV benchmark.

### E.9 RULER

RULER generates synthetic examples to evaluate long-context language models with configurable sequence length and task complexity. RULER includes four task categories, and we select the representative task from each category for our assessment. Following the experimental setup described in Section 4.2, we set the masking group size to 64 and test performance across various context lengths (4k, 8k, 16k, and 31k). We compare CoKV with HeadKV-R2 because HeadKV-R2 is not only the strongest baseline method but also provides per-head importance scores. We use mistral-7B-v0.2-instruct model in this experiment, which supports a 32k-token context window. Notably, the Llama3-8B instruct model is omitted due to its limited 8k context length. As shown in Table 22, our method shows significant advantages over baseline approaches: masking less important groups (low) results in less performance degradation, while masking critical groups (top) leads to substantially larger drops in performance compared with other methods.

Table 13: Benchmark Results of Llama-3-8B-Instruct

| Method | Single-Doc. QA | | | Multi-Doc. QA | | | Summarization | | | Few-shot Learning | | | Synthetic | | Code | | Avg |
|---|---|---|---|---|---|---|---|---|---|---|---|---|---|---|---|---|---|
| | NrtQA | Qasper | MF-en | HotpotQA | 2WikiMQA | Musique | GovReport | QMSum | MultiNews | TREC | TriviaQA | SAMSum | PCount | PRe | Lcc | RB-P | |
| Full Cache | 24.12 | 31.24 | 39.85 | 45.23 | 34.56 | 21.09 | 28.38 | 23.24 | 26.52 | 74.12 | 90.96 | 42.37 | 4.55 | 71.76 | 58.1 | 51.64 | 41.73 |
| **KV size=64** | | | | | | | | | | | | | | | | | |
| SnapKV | 19.94 | 13.21 | 28.91 | 40.06 | 28.58 | 18.12 | 17.29 | 21.71 | 17.05 | 49.41 | 89.00 | 35.48 | 3.99 | 71.57 | 54.35 | 50.42 | 34.94 |
| Pyramid | 20.11 | 16.54 | 32.67 | 40.25 | 27.71 | 17.54 | 18.67 | 22.37 | 20.03 | 62.55 | 89.89 | 36.63 | 4.30 | 71.76 | 54.27 | 50.96 | 36.64 |
| Ada-SnapKV | 20.40 | 14.46 | 32.62 | 42.39 | 31.48 | 17.58 | 18.57 | 22.18 | 18.71 | 58.82 | 90.13 | 35.25 | 4.41 | 71.57 | 54.02 | 51.68 | 36.52 |
| HeadKV-R2 | 20.30 | 16.76 | 35.96 | 38.08 | 26.41 | 17.98 | 18.68 | 21.75 | 20.58 | 67.06 | 88.19 | 37.30 | 3.21 | 71.76 | 56.20 | 54.49 | 37.17 |
| CoKV | 20.77 | 19.67 | 35.11 | 44.37 | 34.36 | 17.83 | 17.89 | 22.33 | 18.55 | 71.76 | 90.73 | 38.51 | 4.71 | 71.76 | 55.45 | 55.82 | 38.73 |
| **KV size=128** | | | | | | | | | | | | | | | | | |
| SnapKV | 20.37 | 14.73 | 34.24 | 43.32 | 28.94 | 19.74 | 19.68 | 22.15 | 20.68 | 64.71 | 90.69 | 39.03 | 4.41 | 71.76 | 58.48 | 51.70 | 37.39 |
| Pyramid | 20.32 | 19.28 | 33.81 | 41.13 | 28.21 | 19.94 | 19.70 | 22.97 | 21.11 | 67.65 | 89.89 | 37.77 | 4.30 | 71.76 | 55.93 | 51.30 | 37.82 |
| Ada-SnapKV | 20.86 | 18.14 | 35.17 | 45.12 | 30.39 | 20.43 | 19.93 | 21.84 | 21.25 | 69.41 | 90.29 | 38.08 | 4.75 | 71.76 | 57.99 | 53.16 | 38.66 |
| HeadKV-R2 | 21.30 | 21.28 | 39.85 | 42.07 | 29.91 | 19.92 | 20.18 | 22.54 | 22.87 | 71.18 | 90.63 | 38.58 | 4.46 | 71.76 | 60.75 | 57.17 | 39.65 |
| CoKV | 20.40 | 23.25 | 38.93 | 45.11 | 37.60 | 20.40 | 19.78 | 23.16 | 21.14 | 73.59 | 91.21 | 40.96 | 4.71 | 71.76 | 58.34 | 59.37 | 40.61 |
| **KV size=256** | | | | | | | | | | | | | | | | | |
| SnapKV | 22.98 | 21.02 | 36.27 | 44.24 | 31.02 | 19.72 | 20.90 | 22.63 | 22.45 | 69.41 | 90.77 | 39.64 | 4.26 | 71.76 | 59.44 | 54.35 | 39.43 |
| Pyramid | 22.18 | 22.83 | 35.95 | 41.85 | 31.74 | 21.14 | 21.27 | 22.65 | 22.83 | 71.18 | 90.83 | 40.50 | 4.35 | 71.37 | 57.69 | 51.49 | 39.37 |
| Ada-SnapKV | 23.58 | 23.76 | 35.65 | 43.83 | 32.24 | 20.50 | 21.26 | 22.77 | 22.69 | 71.76 | 90.87 | 40.36 | 4.21 | 71.76 | 58.79 | 54.70 | 39.92 |
| HeadKV-R2 | 23.13 | 25.55 | 39.97 | 43.60 | 31.12 | 21.26 | 22.02 | 22.68 | 24.47 | 71.76 | 90.63 | 38.32 | 5.13 | 71.08 | 61.81 | 59.25 | 40.74 |
| CoKV | 22.69 | 28.23 | 42.34 | 46.32 | 36.38 | 21.17 | 21.17 | 23.64 | 23.08 | 72.94 | 90.93 | 42.07 | 4.71 | 71.76 | 62.40 | 61.92 | 41.98 |
| **KV size=512** | | | | | | | | | | | | | | | | | |
| SnapKV | 22.92 | 22.86 | 39.33 | 43.89 | 32.70 | 20.87 | 22.24 | 22.39 | 23.97 | 71.18 | 90.87 | 41.14 | 4.54 | 71.76 | 59.98 | 55.00 | 40.35 |
| Pyramid | 23.59 | 25.70 | 38.21 | 44.34 | 32.48 | 20.59 | 22.94 | 22.49 | 24.07 | 72.35 | 90.87 | 40.92 | 4.75 | 71.76 | 58.22 | 52.54 | 40.36 |
| Ada-SnapKV | 23.47 | 28.41 | 39.02 | 44.87 | 32.77 | 20.52 | 23.14 | 22.96 | 24.47 | 72.12 | 90.93 | 39.85 | 4.71 | 71.76 | 58.59 | 54.65 | 40.77 |
| HeadKV-R2 | 22.52 | 29.32 | 40.34 | 45.64 | 34.52 | 20.53 | 23.92 | 22.61 | 25.73 | 72.35 | 90.93 | 39.28 | 4.41 | 71.76 | 61.59 | 59.22 | 41.54 |
| CoKV | 24.56 | 29.18 | 40.60 | 46.11 | 37.53 | 21.33 | 23.02 | 23.51 | 24.77 | 72.94 | 91.09 | 41.29 | 4.76 | 71.50 | 63.06 | 63.55 | 42.44 |
| **KV size=1024** | | | | | | | | | | | | | | | | | |
| SnapKV | 23.95 | 26.95 | 37.81 | 44.03 | 30.88 | 20.93 | 24.26 | 23.09 | 25.79 | 72.35 | 90.87 | 41.43 | 4.31 | 71.76 | 59.29 | 54.91 | 40.79 |
| Pyramid | 23.62 | 26.76 | 39.44 | 45.79 | 33.41 | 19.87 | 23.57 | 22.98 | 25.13 | 73.02 | 90.93 | 40.86 | 4.71 | 71.76 | 58.43 | 53.67 | 40.87 |
| Ada-SnapKV | 23.52 | 28.33 | 40.39 | 45.20 | 32.95 | 20.11 | 24.55 | 23.33 | 25.37 | 73.53 | 90.87 | 41.38 | 4.46 | 71.76 | 58.88 | 54.65 | 41.21 |
| HeadKV-R2 | 23.35 | 29.60 | 40.09 | 45.82 | 35.81 | 21.39 | 25.57 | 23.32 | 26.30 | 74.12 | 90.77 | 40.27 | 4.19 | 71.76 | 61.58 | 59.03 | 42.06 |
| CoKV | 24.01 | 31.70 | 40.64 | 48.13 | 37.89 | 20.64 | 23.02 | 23.89 | 25.71 | 74.12 | 91.01 | 42.02 | 4.71 | 71.20 | 63.33 | 63.74 | 42.86 |

Table 14: Benchmark Results of Qwen3-32B

| Method | Single-Doc. QA | | | Multi-Doc. QA | | | Summarization | | | Few-shot Learning | | | Synthetic | | Code | | Avg |
|---|---|---|---|---|---|---|---|---|---|---|---|---|---|---|---|---|---|
| | NrtQA | Qasper | MF-en | HotpotQA | 2WikiMQA | Musique | GovReport | QMSum | MultiNews | TREC | TriviaQA | SAMSum | PCount | PRe | Lcc | RB-P | |
| Full Cache | 37.14 | 45.51 | 49.17 | 58.2 | 54.74 | 38.36 | 32.9 | 23.72 | 25.08 | 72.78 | 72.57 | 37.95 | 17.78 | 100 | 62.72 | 70.08 | 49.92 |
| **KV size=64** | | | | | | | | | | | | | | | | | |
| SnapKV | 28.97 | 30.05 | 41.1 | 29.74 | 41.62 | 35.03 | 16.57 | 19.56 | 16.86 | 33.33 | 67.43 | 32.3 | 9.52 | 95.24 | 50.67 | 47.05 | 37.19 |
| Pyramid | 19.74 | 30.56 | 36.64 | 32.77 | 40.62 | 35.56 | 17.62 | 21.29 | 16.53 | 33.33 | 67.43 | 36.61 | 9.52 | 100.0 | 55.33 | 48.95 | 37.66 |
| Ada-SnapKV | 31.71 | 32.31 | 37.52 | 32.72 | 42.53 | 37.32 | 17.17 | 21.89 | 17.1 | 33.33 | 71.43 | 32.35 | 9.52 | 100.0 | 53.24 | 47.19 | 38.58 |
| HeadKV-R2 | 29.1 | 32.46 | 39.89 | 50.95 | 44.62 | 33.42 | 18.04 | 20.01 | 17.27 | 41.41 | 66.06 | 32.37 | 18.2 | 98.99 | 55.0 | 50.87 | 40.54 |
| CoKV | 29.37 | 33.52 | 43.15 | 52.83 | 48.37 | 37.1 | 18.18 | 19.96 | 18.72 | 44.44 | 68.77 | 32.67 | 19.21 | 98.99 | 55.08 | 51.4 | 41.99 |
| **KV size=128** | | | | | | | | | | | | | | | | | |
| SnapKV | 30.18 | 32.54 | 41.96 | 55.2 | 47.04 | 34.31 | 22.74 | 21.19 | 19.0 | 47.22 | 67.89 | 36.77 | 17.22 | 98.89 | 58.52 | 50.48 | 42.65 |
| Pyramid | 27.17 | 32.11 | 40.93 | 30.13 | 37.2 | 34.43 | 22.01 | 20.95 | 18.71 | 38.1 | 68.43 | 35.55 | 9.52 | 100.0 | 56.48 | 51.29 | 40.19 |
| Ada-SnapKV | 22.74 | 32.11 | 40.02 | 32.42 | 40.44 | 27.97 | 23.43 | 21.67 | 18.98 | 46.67 | 67.43 | 38.93 | 9.52 | 100.0 | 57.9 | 49.71 | 39.43 |
| HeadKV-R2 | 29.78 | 33.39 | 41.51 | 51.85 | 52.06 | 36.17 | 24.17 | 21.04 | 19.03 | 48.89 | 67.98 | 36.44 | 18.02 | 99.44 | 56.88 | 52.47 | 43.07 |
| CoKV | 28.73 | 37.56 | 44.80 | 53.48 | 53.07 | 35.17 | 23.89 | 21.34 | 19.28 | 54.44 | 69.35 | 38.23 | 19.22 | 100 | 57.13 | 56.87 | 44.53 |
| **KV size=256** | | | | | | | | | | | | | | | | | |
| SnapKV | 32.96 | 37.46 | 46.19 | 38.2 | 47.63 | 38.21 | 25.04 | 22.29 | 20.88 | 42.86 | 68.43 | 44.41 | 9.52 | 100.0 | 62.05 | 52.62 | 43.05 |
| Pyramid | 29.13 | 31.77 | 47.74 | 31.19 | 40.27 | 26.84 | 24.97 | 22.75 | 20.97 | 38.1 | 68.43 | 37.85 | 9.52 | 100.0 | 58.38 | 55.0 | 40.18 |
| Ada-SnapKV | 30.26 | 37.95 | 48.05 | 37.54 | 48.17 | 42.59 | 25.33 | 23.9 | 21.15 | 47.62 | 68.93 | 41.95 | 9.52 | 100.0 | 64.0 | 53.14 | 43.76 |
| HeadKV-R2 | 33.0 | 38.69 | 47.01 | 51.98 | 57.2 | 38.36 | 26.01 | 22.17 | 21.83 | 49.49 | 68.15 | 38.79 | 18.18 | 100.0 | 58.35 | 55.72 | 45.31 |
| CoKV | 33.07 | 41.66 | 47.5 | 55.98 | 59.15 | 38.88 | 25.84 | 22.79 | 21.41 | 56.57 | 68.98 | 38.65 | 17.17 | 100.0 | 59.42 | 55.76 | 46.43 |
| **KV size=1024** | | | | | | | | | | | | | | | | | |
| SnapKV | 33.75 | 43.59 | 46.97 | 45.74 | 56.52 | 38.19 | 28.10 | 22.96 | 24.18 | 61.9 | 69.25 | 45.12 | 9.52 | 100.0 | 60.86 | 67.1 | 47.11 |
| Ada-SnapKV | 34.92 | 44.87 | 47.18 | 46.45 | 56.21 | 37.19 | 27.74 | 23.56 | 24.42 | 61.9 | 69.52 | 40.54 | 9.52 | 100.0 | 61.71 | 63.38 | 46.82 |
| HeadKV-R2 | 36.27 | 44.14 | 50.83 | 54.07 | 62.96 | 37.72 | 28.29 | 23.79 | 24.63 | 64.65 | 68.48 | 39.07 | 18.28 | 100.0 | 61.29 | 64.91 | 48.71 |
| CoKV | 35.82 | 45.23 | 51.45 | 57.31 | 61.37 | 37.34 | 28.78 | 24.41 | 24.79 | 68.69 | 68.96 | 39.01 | 18.18 | 100.0 | 62.52 | 65.62 | 49.34 |

Table 15: Results of Mistral-7B-Instruct-v0.2

| Method | Single-Doc. QA | | | Multi-Doc. QA | | | Summarization | | | Few-shot Learning | | | Synthetic | | Code | | Avg |
| --- | --- | --- | --- | --- | --- | --- | --- | --- | --- | --- | --- | --- | --- | --- | --- | --- | --- |
| | NrtQA | Qasper | MF-en | HotpotQA | 2WikiMQA | Musique | GovReport | QMSum | MultiNews | TREC | TriviaQA | SAMSum | PCount | PRe | Lcc | RB-P | |
| Full Cache | 26.40 | 31.07 | 49.38 | 37.60 | 26.07 | 17.81 | 31.87 | 23.16 | 27.15 | 70.59 | 85.73 | 43.26 | 1.52 | 58.52 | 55.10 | 49.45 | 39.67 |
| **KV size=64** | | | | | | | | | | | | | | | | | |
| SnapKV | 16.99 | 18.26 | 38.29 | 29.51 | 23.24 | 13.46 | 18.24 | 20.48 | 18.05 | 48.82 | 81.45 | 36.18 | 2.54 | 43.79 | 46.13 | 39.30 | 30.92 |
| Pyramid | 17.51 | 18.60 | 40.49 | 31.92 | 22.08 | 13.81 | 18.68 | 20.94 | 18.80 | 57.06 | 81.71 | 37.42 | 1.68 | 46.23 | 46.05 | 40.03 | 32.06 |
| Ada-SnapKV | 17.93 | 18.68 | 40.03 | 29.99 | 22.67 | 14.92 | 18.84 | 20.87 | 18.53 | 54.12 | 81.43 | 37.25 | 2.30 | 45.20 | 46.84 | 39.37 | 31.81 |
| HeadKV-R2 | 22.75 | 25.37 | 45.36 | 36.52 | 25.39 | 13.82 | 20.45 | 22.06 | 21.48 | 65.29 | 83.56 | 37.95 | 2.43 | 50.78 | 47.76 | 42.86 | 35.24 |
| CoKV | 21.07 | 21.41 | 42.87 | 37.74 | 28.93 | 15.60 | 18.03 | 21.08 | 19.70 | 67.65 | 86.52 | 39.54 | 3.68 | 54.22 | 49.20 | 42.13 | 35.59 |
| **KV size=128** | | | | | | | | | | | | | | | | | |
| SnapKV | 23.02 | 20.73 | 41.91 | 31.39 | 22.88 | 14.55 | 20.92 | 21.83 | 21.25 | 62.35 | 83.21 | 38.99 | 3.14 | 51.16 | 49.94 | 43.61 | 34.43 |
| Pyramid | 22.06 | 21.82 | 43.73 | 32.33 | 24.12 | 13.80 | 20.27 | 21.65 | 21.34 | 65.29 | 83.78 | 38.37 | 2.63 | 53.59 | 49.21 | 42.69 | 34.79 |
| Ada-SnapKV | 22.32 | 22.71 | 44.40 | 32.63 | 23.29 | 13.79 | 21.15 | 22.50 | 21.77 | 66.47 | 84.28 | 39.68 | 3.04 | 51.87 | 49.57 | 44.84 | 35.27 |
| HeadKV-R2 | 24.81 | 27.66 | 48.29 | 36.87 | 26.66 | 14.75 | 23.30 | 22.88 | 23.26 | 67.65 | 84.93 | 39.75 | 2.50 | 49.31 | 50.79 | 45.57 | 36.81 |
| CoKV | 24.42 | 24.12 | 46.95 | 38.28 | 28.85 | 17.18 | 21.11 | 21.91 | 22.02 | 68.82 | 86.14 | 40.48 | 4.21 | 54.12 | 51.08 | 46.25 | 37.25 |
| **KV size=256** | | | | | | | | | | | | | | | | | |
| SnapKV | 23.01 | 23.47 | 45.38 | 33.15 | 24.12 | 13.93 | 22.80 | 22.89 | 22.85 | 67.65 | 84.62 | 40.39 | 2.36 | 59.18 | 51.34 | 46.74 | 36.49 |
| Pyramid | 22.98 | 25.66 | 46.12 | 34.47 | 25.81 | 13.98 | 22.86 | 22.54 | 22.88 | 68.90 | 85.07 | 40.92 | 2.39 | 58.74 | 53.13 | 46.59 | 37.07 |
| Ada-SnapKV | 23.54 | 26.02 | 45.92 | 34.45 | 26.09 | 14.12 | 22.79 | 22.64 | 23.32 | 68.82 | 85.32 | 41.93 | 2.04 | 58.62 | 52.10 | 47.70 | 37.21 |
| HeadKV-R2 | 25.40 | 27.42 | 47.05 | 37.98 | 25.57 | 17.08 | 25.31 | 22.72 | 25.03 | 69.41 | 84.93 | 40.24 | 2.58 | 52.94 | 53.48 | 49.21 | 37.90 |
| CoKV | 25.70 | 26.10 | 48.43 | 38.96 | 30.06 | 17.33 | 23.42 | 22.55 | 23.73 | 70.00 | 86.19 | 42.35 | 3.65 | 56.37 | 53.97 | 48.79 | 38.60 |
| **KV size=512** | | | | | | | | | | | | | | | | | |
| SnapKV | 25.24 | 26.30 | 47.85 | 37.16 | 25.07 | 14.57 | 24.43 | 22.98 | 24.61 | 68.82 | 85.72 | 43.04 | 2.00 | 58.63 | 54.06 | 49.03 | 38.09 |
| Pyramid | 24.43 | 27.09 | 48.49 | 37.57 | 25.35 | 16.20 | 24.40 | 22.85 | 24.16 | 68.82 | 85.81 | 42.07 | 1.87 | 56.93 | 53.05 | 48.22 | 37.96 |
| Ada-SnapKV | 25.01 | 26.76 | 49.10 | 37.12 | 26.68 | 15.63 | 24.42 | 22.94 | 24.61 | 69.41 | 85.56 | 41.88 | 1.87 | 57.93 | 54.09 | 48.94 | 38.25 |
| HeadKV-R2 | 25.80 | 28.73 | 48.34 | 37.43 | 27.03 | 17.28 | 28.22 | 23.22 | 25.50 | 70.59 | 85.72 | 40.15 | 2.69 | 56.15 | 53.24 | 49.22 | 38.78 |
| CoKV | 25.25 | 28.13 | 49.91 | 38.87 | 32.33 | 18.27 | 25.00 | 23.08 | 25.50 | 70.59 | 86.37 | 43.46 | 3.06 | 59.20 | 55.54 | 49.38 | 39.62 |
| **KV size=1024** | | | | | | | | | | | | | | | | | |
| SnapKV | 26.38 | 29.70 | 48.13 | 37.36 | 25.52 | 16.88 | 27.31 | 22.63 | 26.10 | 69.41 | 85.72 | 42.43 | 1.54 | 56.87 | 55.05 | 49.33 | 38.77 |
| Pyramid | 25.09 | 28.59 | 47.78 | 37.74 | 25.83 | 17.53 | 25.88 | 23.05 | 25.91 | 68.24 | 85.95 | 42.77 | 1.59 | 57.82 | 54.47 | 48.85 | 38.57 |
| Ada-SnapKV | 25.70 | 29.95 | 47.50 | 37.68 | 26.18 | 17.10 | 26.63 | 22.93 | 26.10 | 70.00 | 85.72 | 43.16 | 1.68 | 56.28 | 54.52 | 49.10 | 38.76 |
| HeadKV-R2 | 27.48 | 29.94 | 49.49 | 37.49 | 26.45 | 18.69 | 30.73 | 23.31 | 26.74 | 70.59 | 85.92 | 42.05 | 3.15 | 56.37 | 54.73 | 49.30 | 39.53 |
| CoKV | 26.15 | 29.82 | 49.47 | 38.54 | 34.39 | 17.98 | 27.76 | 23.33 | 26.49 | 70.59 | 86.23 | 43.54 | 2.48 | 59.32 | 55.47 | 49.92 | 40.09 |

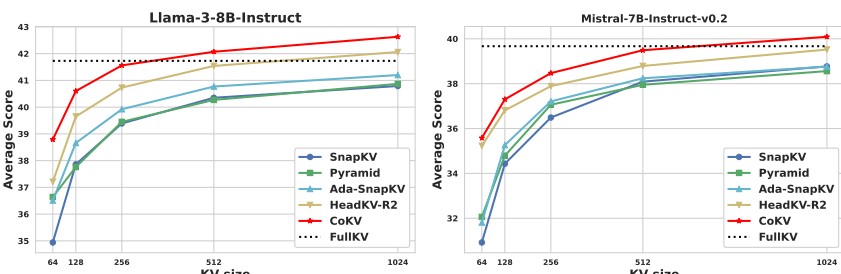

Figure 4: Results for varying KV cache sizes (64, 128, 256, 512, 1024) in the LongBench benchmark.

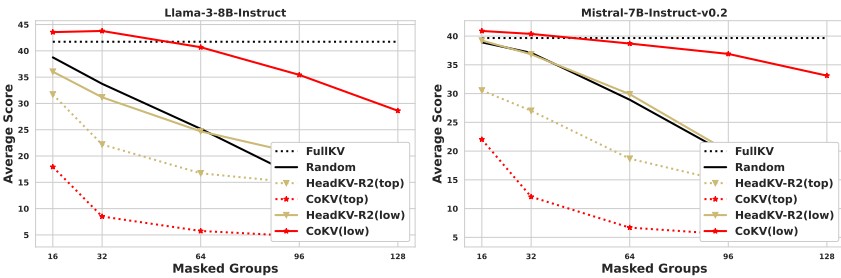

Figure 5: Results for varying masked groups (16,32,64,96,128) in the LongBench benchmark.

Table 16: Results of masking groups with Llama-3-8B-Instruct

| Method | Single-Doc. QA | | | Multi-Doc. QA | | | Summarization | | | Few-shot Learning | | | Synthetic | | Code | | Avg |
|---|---|---|---|---|---|---|---|---|---|---|---|---|---|---|---|---|---|
| | NtrQA | Qasper | MF-en | HotpotQA | 2WikiMQA | Musique | GovReport | QMSum | MultiNews | TREC | TriviaQA | SAMSum | PCount | PRe | Lcc | RB-P | |
| Full Cache | 24.12 | 31.24 | 39.85 | 45.23 | 34.56 | 21.09 | 28.38 | 23.24 | 26.52 | 74.12 | 90.96 | 42.37 | 4.55 | 71.76 | 58.10 | 51.64 | 41.73 |
| **Masking 16 groups** | | | | | | | | | | | | | | | | | |
| Random | 20.93 | 28.48 | 33.69 | 44.93 | 20.01 | 20.6 | 28.43 | 23.7 | 26.67 | 74.12 | 91.07 | 41.12 | 4.26 | 71.76 | 49.83 | 40.55 | 38.76 |
| HeadKV-R2(top) | 19.45 | 12.97 | 27.75 | 34.2 | 17.33 | 14.32 | 19.74 | 22.76 | 22.05 | 67.06 | 87.91 | 35.53 | 4.71 | 68.49 | 26.62 | 26.53 | 31.71 |
| CoKV(top) | 6.55 | 9.46 | 9.47 | 10.19 | 12.27 | 5.67 | 5.73 | 16.96 | 4.47 | 43.53 | 71.21 | 23.77 | 3.91 | 34.98 | 11.58 | 17.18 | 17.93 |
| HeadKV-R2(low) | 21.83 | 14.36 | 33.34 | 31.37 | 27.23 | 12.55 | 27.29 | 23.82 | 26.99 | 74.12 | 91.03 | 42.18 | 4.12 | 70.59 | 37.35 | 38.55 | 36.05 |
| CoKV(low) | 23.74 | 33.76 | 41.71 | 49.27 | 40.48 | 19.99 | 29.13 | 23.25 | 27.79 | 74.12 | 91.45 | 42.37 | 4.71 | 70.55 | 63.38 | 61.26 | 43.56 |
| **Masking 32 groups** | | | | | | | | | | | | | | | | | |
| Random | 20.69 | 18.60 | 29.63 | 39.12 | 18.50 | 6.94 | 22.40 | 22.33 | 26.45 | 74.12 | 89.82 | 33.80 | 4.71 | 61.12 | 30.78 | 40.71 | 33.73 |
| HeadKV-R2(top) | 17.33 | 6.98 | 9.37 | 13.50 | 9.37 | 5.11 | 13.18 | 20.86 | 15.24 | 45.88 | 75.30 | 27.21 | 4.76 | 66.21 | 11.24 | 13.64 | 22.20 |
| CoKV(top) | 1.40 | 3.49 | 3.78 | 7.94 | 9.32 | 2.32 | 2.64 | 11.74 | 0.58 | 34.71 | 21.37 | 6.96 | 4.14 | 16.93 | 3.54 | 5.17 | 8.50 |
| HeadKV-R2(low) | 21.51 | 11.16 | 25.33 | 19.52 | 14.48 | 7.42 | 16.73 | 23.91 | 14.58 | 74.12 | 89.09 | 40.69 | 4.66 | 70.09 | 33.13 | 32.39 | 31.18 |
| CoKV(low) | 22.45 | 33.06 | 38.34 | 45.82 | 39.62 | 20.18 | 28.39 | 24.04 | 26.67 | 74.12 | 91.14 | 41.70 | 4.71 | 71.76 | 52.24 | 64.94 | 42.45 |
| **Masking 64 groups** | | | | | | | | | | | | | | | | | |
| Random | 13.22 | 7.34 | 20.57 | 20.58 | 9.11 | 6.76 | 7.50 | 21.22 | 19.18 | 72.35 | 71.92 | 36.09 | 4.71 | 52.80 | 21.27 | 18.07 | 25.17 |
| HeadKV-R2(top) | 7.49 | 2.95 | 5.05 | 11.06 | 12.01 | 2.46 | 3.63 | 14.43 | 5.06 | 34.71 | 48.92 | 8.05 | 3.97 | 70.67 | 21.03 | 16.14 | 16.73 |
| CoKV (top) | 0.76 | 1.76 | 2.45 | 4.85 | 5.58 | 1.93 | 2.48 | 5.65 | 0.20 | 34.12 | 3.33 | 7.34 | 3.16 | 12.18 | 2.45 | 3.83 | 5.75 |
| HeadKV-R2(low) | 19.23 | 12.19 | 21.33 | 19.61 | 14.21 | 6.63 | 6.45 | 20.17 | 6.16 | 71.76 | 77.40 | 31.52 | 4.41 | 53.48 | 16.00 | 14.58 | 24.70 |
| CoKV(low) | 21.98 | 29.85 | 38.95 | 44.21 | 36.65 | 17.71 | 28.04 | 24.49 | 25.92 | 74.71 | 91.66 | 40.80 | 4.54 | 71.76 | 47.04 | 52.77 | 40.69 |
| **Masking 96 groups** | | | | | | | | | | | | | | | | | |
| Random | 5.19 | 4.04 | 6.85 | 8.15 | 10.33 | 5.08 | 2.21 | 10.77 | 2.82 | 40.00 | 61.54 | 13.38 | 4.64 | 54.29 | 15.37 | 9.81 | 15.90 |
| HeadKV-R2(top) | 2.89 | 4.34 | 7.90 | 11.83 | 9.14 | 2.93 | 4.37 | 13.21 | 3.80 | 34.12 | 30.32 | 8.46 | 4.78 | 71.76 | 13.55 | 14.76 | 14.89 |
| CoKV(top) | 1.36 | 1.14 | 1.82 | 3.66 | 3.79 | 1.48 | 1.20 | 4.63 | 0.13 | 34.12 | 2.40 | 7.52 | 0.54 | 6.71 | 2.41 | 3.54 | 4.78 |
| HeadKV-R2(low) | 19.28 | 8.23 | 15.65 | 20.89 | 16.80 | 8.00 | 3.32 | 11.81 | 0.99 | 58.82 | 58.70 | 15.72 | 4.71 | 61.88 | 10.56 | 11.05 | 20.40 |
| CoKV(low) | 20.24 | 18.97 | 35.28 | 41.37 | 30.02 | 13.87 | 19.95 | 17.33 | 20.76 | 74.71 | 84.08 | 41.23 | 4.71 | 68.24 | 38.11 | 38.08 | 35.43 |
| **Masking 128 groups** | | | | | | | | | | | | | | | | | |
| Random | 3.34 | 2.50 | 5.33 | 10.59 | 5.12 | 2.73 | 2.15 | 9.19 | 0.16 | 44.12 | 31.33 | 9.05 | 4.18 | 66.74 | 12.27 | 9.23 | 13.63 |
| HeadKV-R2(top) | 2.34 | 2.17 | 5.38 | 7.21 | 7.19 | 1.85 | 1.80 | 10.34 | 0.31 | 34.71 | 26.08 | 7.87 | 4.71 | 66.92 | 13.94 | 11.76 | 12.79 |
| CoKV(top) | 0.59 | 0.80 | 1.38 | 2.96 | 3.42 | 1.11 | 1.16 | 4.05 | 0.13 | 34.12 | 2.89 | 7.17 | 1.09 | 7.52 | 2.91 | 3.55 | 4.68 |
| HeadKV-R2(low) | 12.02 | 7.97 | 8.92 | 14.87 | 12.83 | 5.26 | 2.41 | 9.12 | 1.42 | 55.88 | 40.96 | 10.2 | 4.71 | 68.42 | 10.14 | 6.03 | 16.95 |
| CoKV(low) | 15.31 | 12.15 | 28.44 | 35.35 | 23.27 | 10.67 | 2.93 | 12.24 | 9.41 | 73.82 | 76.32 | 37.70 | 4.71 | 68.24 | 22.20 | 24.93 | 28.61 |

Table 17: Results of masking groups with Qwen3-32B

| Method | Single-Doc. QA | | | Multi-Doc. QA | | | Summarization | | | Few-shot Learning | | | Synthetic | | Code | | Avg |
|---|---|---|---|---|---|---|---|---|---|---|---|---|---|---|---|---|---|
| | NtrQA | Qasper | MF-en | HotpotQA | 2WikiMQA | Musique | GovReport | QMSum | MultiNews | TREC | TriviaQA | SAMSum | PCount | PRe | Lcc | RB-P | |
| Full Cache | 37.14 | 45.51 | 49.17 | 58.2 | 54.74 | 38.36 | 32.9 | 23.72 | 25.08 | 72.78 | 72.57 | 37.95 | 17.78 | 100 | 62.72 | 70.08 | 49.92 |
| **Masking 16 groups** | | | | | | | | | | | | | | | | | |
| Random | 36.53 | 45.37 | 54.32 | 55.82 | 61.69 | 40.09 | 26.43 | 24.57 | 25.04 | 69.7 | 68.57 | 37.43 | 22.22 | 100.0 | 22.51 | 25.01 | 44.71 |
| HeadKV-R2(top) | 36.89 | 45.22 | 54.08 | 56.57 | 61.02 | 38.12 | 24.08 | 22.58 | 24.69 | 68.43 | 68.43 | 35.28 | 19.19 | 57.05 | 26.50 | 28.61 | 41.67 |
| CoKV(top) | 32.16 | 35.64 | 38.05 | 35.18 | 29.20 | 30.56 | 21.38 | 23.92 | 24.67 | 38.61 | 37.57 | 27.02 | 19.57 | 35.93 | 9.74 | 12.43 | 28.23 |
| CoKV(top)-general | 23.7 | 31.61 | 28.7 | 32.91 | 21.91 | 15.86 | 22.12 | 20.73 | 13.58 | 40.4 | 39.41 | 24.96 | 19.19 | 15.15 | 16.07 | 17.75 | 24.00 |
| **Masking 32 groups** | | | | | | | | | | | | | | | | | |
| Random | 33.75 | 43.74 | 54.44 | 53.97 | 63.31 | 39.09 | 25.01 | 24.86 | 25.31 | 69.7 | 67.56 | 35.92 | 21.21 | 100.0 | 18.11 | 25.76 | 43.86 |
| HeadKV-R2(top) | 36.52 | 46.09 | 53.75 | 56.14 | 60.98 | 38.35 | 21.96 | 23.96 | 24.67 | 68.69 | 63.8 | 34.09 | 21.21 | 100.0 | 21.13 | 22.77 | 43.38 |
| CoKV(top) | 35.42 | 33.8 | 27.53 | 32.42 | 24.45 | 22.63 | 18.67 | 24.13 | 25.0 | 36.36 | 29.67 | 20.87 | 19.19 | 29.29 | 5.18 | 6.89 | 24.47 |
| CoKV(top)-general | 17.59 | 23.02 | 13.13 | 22.35 | 16.09 | 8.1 | 19.03 | 16.08 | 9.91 | 36.36 | 26.18 | 15.34 | 20.2 | 41.41 | 10.86 | 16.55 | 19.51 |
| **Masking 64 groups** | | | | | | | | | | | | | | | | | |
| Random | 30.13 | 44.81 | 50.07 | 56.51 | 54.06 | 39.24 | 27.33 | 23.32 | 24.82 | 72.78 | 71.12 | 37.72 | 12.94 | 100.0 | 22.71 | 23.79 | 43.21 |
| HeadKV-R2(top) | 28.38 | 34.55 | 32.15 | 47.25 | 45.26 | 25.33 | 20.14 | 22.17 | 13.98 | 55.56 | 56.69 | 21.18 | 8.69 | 98.33 | 14.44 | 18.47 | 33.91 |
| CoKV(top) | 28.85 | 28.55 | 18.91 | 25.1 | 19.78 | 12.69 | 13.48 | 22.97 | 25.0 | 35.56 | 26.43 | 10.41 | 8.04 | 17.22 | 5.2 | 5.05 | 18.88 |
| CoKV(top)-general | 12.07 | 16.96 | 7.72 | 19.81 | 16.13 | 4.72 | 14.47 | 12.14 | 4.72 | 34.34 | 22.01 | 8.58 | 14.14 | 1.01 | 15.76 | 14.37 | 13.68 |
| **Masking 96 groups** | | | | | | | | | | | | | | | | | |
| Random | 36.07 | 45.2 | 54.41 | 57.49 | 60.17 | 40.92 | 22.67 | 24.37 | 24.5 | 68.69 | 63.46 | 21.53 | 21.21 | 96.97 | 24.15 | 23.12 | 42.81 |
| HeadKV-R2(top) | 22.16 | 23.92 | 21.57 | 38.2 | 38.23 | 21.34 | 15.85 | 17.13 | 7.72 | 38.38 | 42.65 | 11.54 | 6.23 | 46.46 | 6.23 | 10.23 | 23.86 |
| CoKV(top) | 24.3 | 22.02 | 13.6 | 18.44 | 14.94 | 13.98 | 11.39 | 22.44 | 16.66 | 33.33 | 17.8 | 6.23 | 13.13 | 3.03 | 5.28 | 4.51 | 15.07 |
| CoKV(top)-general | 8.4 | 11.81 | 7.76 | 15.6 | 12.53 | 6.97 | 9.82 | 9.74 | 3.4 | 33.33 | 21.65 | 8.71 | 14.14 | 0.0 | 13.68 | 9.25 | 11.67 |

Table 18: Results of masking groups with Mistral-7B-Instruct-v0.2

| Method | Single-Doc. QA | | | Multi-Doc. QA | | | Summarization | | | Few-shot Learning | | | Synthetic | | Code | | Avg |
|---|---|---|---|---|---|---|---|---|---|---|---|---|---|---|---|---|---|
| | NrtQA | Qasper | MF-en | HotpotQA | 2WikiMQA | Musique | GovReport | QMSum | MultiNews | TREC | TriviaQA | SAMSum | PCount | PRe | Lcc | RB-P | |
| Full Cache | 26.40 | 31.07 | 49.38 | 37.60 | 26.07 | 17.81 | 31.87 | 23.16 | 27.15 | 70.59 | 85.73 | 43.26 | 1.52 | 58.52 | 55.10 | 49.45 | 39.67 |
| **Masking 16 groups** | | | | | | | | | | | | | | | | | |
| Random | 25.92 | 31.73 | 50.29 | 37.84 | 27.19 | 17.83 | 24.91 | 21.92 | 27.04 | 70.59 | 85.93 | 43.8 | 3.22 | 53.82 | 52.38 | 48.24 | 38.92 |
| HeadKV-R2(top) | 23.38 | 16.66 | 37.13 | 37.41 | 22.76 | 14.29 | 18.8 | 21.74 | 23.23 | 54.12 | 82.96 | 35.22 | 4.12 | 21.76 | 39.49 | 35.66 | 30.55 |
| CoKV(top) | 16.1 | 23.35 | 18.49 | 14.34 | 13.39 | 7.89 | 20.5 | 19.98 | 17.25 | 38.24 | 52.51 | 26.32 | 4.17 | 40.85 | 24.6 | 14.35 | 22.02 |
| HeadKV-R2(low) | 24.78 | 29.37 | 48.78 | 38.07 | 24.88 | 16.93 | 31.25 | 23.08 | 27.64 | 71.18 | 84.55 | 42.52 | 2.1 | 58.82 | 54.22 | 49.4 | 39.22 |
| CoKV(low) | 26.57 | 32.3 | 49.94 | 40.38 | 34.0 | 19.11 | 31.25 | 22.97 | 26.85 | 70.59 | 87.3 | 44.39 | 3.29 | 58.03 | 56.6 | 50.74 | 40.89 |
| **Masking 32 groups** | | | | | | | | | | | | | | | | | |
| Random | 22.62 | 31.72 | 47.20 | 38.13 | 22.55 | 11.92 | 25.64 | 23.27 | 26.75 | 68.82 | 84.55 | 41.34 | 1.93 | 49.71 | 50.14 | 47.18 | 37.09 |
| HeadKV-R2(top) | 20.82 | 15.40 | 28.72 | 34.31 | 20.31 | 12.86 | 13.56 | 19.83 | 17.80 | 46.47 | 79.25 | 30.10 | 4.71 | 24.31 | 33.41 | 30.47 | 27.02 |
| CoKV(top) | 9.05 | 15.38 | 7.61 | 9.88 | 8.07 | 6.38 | 0.59 | 11.72 | 4.70 | 35.88 | 26.87 | 11.85 | 4.65 | 10.88 | 15.23 | 11.14 | 11.87 |
| HeadKV-R2(low) | 23.76 | 27.40 | 44.80 | 32.85 | 23.55 | 13.28 | 24.37 | 22.71 | 28.09 | 71.18 | 79.24 | 42.24 | 4.26 | 49.90 | 52.89 | 48.85 | 36.84 |
| CoKV(low) | 26.70 | 30.44 | 49.57 | 40.41 | 32.28 | 18.33 | 30.26 | 23.27 | 26.85 | 70.59 | 87.48 | 44.04 | 2.93 | 56.27 | 56.34 | 50.38 | 40.38 |
| **Masking 64 groups** | | | | | | | | | | | | | | | | | |
| Random | 13.43 | 24.46 | 30.97 | 22.62 | 16.93 | 15.65 | 14.07 | 22.16 | 19.86 | 55.29 | 82.16 | 35.85 | 4.12 | 38.94 | 38.07 | 28.39 | 28.94 |
| HeadKV-R2(top) | 11.04 | 9.09 | 17.45 | 18.57 | 13.79 | 8.07 | 9.83 | 17.30 | 12.60 | 35.29 | 55.36 | 18.65 | 4.54 | 19.85 | 26.25 | 21.23 | 18.68 |
| CoKV(top) | 3.28 | 3.50 | 4.65 | 4.30 | 3.42 | 2.55 | 0.79 | 4.66 | 1.08 | 34.71 | 8.41 | 6.00 | 3.53 | 3.53 | 11.22 | 11.57 | 6.70 |
| HeadKV-R2(low) | 18.81 | 21.42 | 35.18 | 18.03 | 14.26 | 7.41 | 22.56 | 22.41 | 20.24 | 57.65 | 75.72 | 37.03 | 4.11 | 45.46 | 38.78 | 39.22 | 29.89 |
| CoKV(low) | 26.87 | 25.74 | 48.19 | 39.61 | 30.86 | 16.88 | 24.45 | 22.84 | 27.29 | 71.18 | 87.16 | 43.43 | 3.34 | 50.18 | 53.76 | 47.52 | 38.71 |
| **Masking 96 groups** | | | | | | | | | | | | | | | | | |
| Random | 4.84 | 6.33 | 13.77 | 12.00 | 10.41 | 8.43 | 0.88 | 17.55 | 21.83 | 51.76 | 63.48 | 22.32 | 4.47 | 34.19 | 21.30 | 17.65 | 19.45 |
| HeadKV-R2(top) | 9.21 | 7.05 | 11.34 | 13.30 | 14.22 | 3.99 | 7.67 | 15.43 | 8.84 | 34.71 | 29.87 | 9.97 | 4.44 | 30.16 | 17.73 | 16.24 | 14.64 |
| CoKV(top) | 2.13 | 4.13 | 4.58 | 4.09 | 6.52 | 0.64 | 0.00 | 2.44 | 0.15 | 34.71 | 2.16 | 4.40 | 4.12 | 2.94 | 7.16 | 8.39 | 5.54 |
| HeadKV-R2(low) | 8.17 | 10.62 | 18.76 | 13.07 | 10.10 | 5.44 | 3.75 | 19.42 | 6.51 | 46.47 | 50.84 | 23.98 | 4.57 | 29.89 | 34.95 | 32.57 | 19.94 |
| CoKV(low) | 24.62 | 24.71 | 48.04 | 38.72 | 30.29 | 16.37 | 19.35 | 22.84 | 27.18 | 70.59 | 79.48 | 42.01 | 3.75 | 48.29 | 50.78 | 43.53 | 36.91 |
| **Masking 128 groups** | | | | | | | | | | | | | | | | | |
| Random | 4.15 | 8.45 | 9.73 | 8.38 | 7.80 | 2.07 | 0.51 | 13.19 | 3.40 | 42.94 | 34.04 | 8.82 | 3.85 | 3.53 | 23.74 | 18.34 | 12.06 |
| HeadKV-R2(top) | 5.22 | 4.78 | 8.63 | 7.04 | 6.15 | 3.89 | 5.64 | 14.59 | 5.64 | 35.88 | 25.98 | 8.36 | 3.82 | 18.53 | 18.68 | 18.52 | 11.96 |
| CoKV(top) | 1.33 | 9.43 | 1.03 | 4.24 | 5.54 | 1.41 | 0.09 | 0.78 | 0.01 | 33.53 | 1.06 | 4.50 | 2.94 | 2.94 | 6.94 | 6.22 | 5.12 |
| HeadKV-R2(low) | 4.41 | 4.53 | 11.12 | 12.8 | 7.20 | 6.64 | 0.46 | 10.48 | 0.61 | 47.65 | 31.61 | 10.45 | 2.91 | 9.92 | 24.09 | 24.48 | 13.09 |
| CoKV(low) | 20.43 | 19.12 | 44.82 | 34.23 | 23.31 | 13.97 | 14.22 | 21.28 | 24.65 | 70.59 | 73.98 | 39.73 | 4.10 | 45.21 | 42.14 | 38.14 | 33.12 |

Table 19: Comprehensive KV Quantization Results on LongBench

| Method | Single-Doc. QA | | | Multi-Doc. QA | | | Summarization | | | Few-shot Learning | | | Synthetic | | Code | | Avg |
|---|---|---|---|---|---|---|---|---|---|---|---|---|---|---|---|---|---|
| | NrtQA | Qasper | MF-en | HotpotQA | 2WikiMQA | Musique | GovReport | QMSum | MultiNews | TREC | TriviaQA | SAMSum | PCount | PRe | Lcc | RB-P | |
| **Qwen3-8B model with 8-4 bits quantization** | | | | | | | | | | | | | | | | | |
| HeadKV-R2 | 10.78 | 33.24 | 32.78 | 29.37 | 25.96 | 7.93 | 13.17 | 28.49 | 60.28 | 77.06 | 51.97 | 26.1 | 9.88 | 79.41 | 60.29 | 55.47 | 37.64 |
| CoKV | 20.43 | 41.36 | 37.74 | 48.44 | 27.54 | 16.53 | 14.22 | 36.58 | 60.18 | 72.94 | 86.76 | 34.12 | 13.51 | 80.59 | 66.26 | 61.7 | 44.93 |

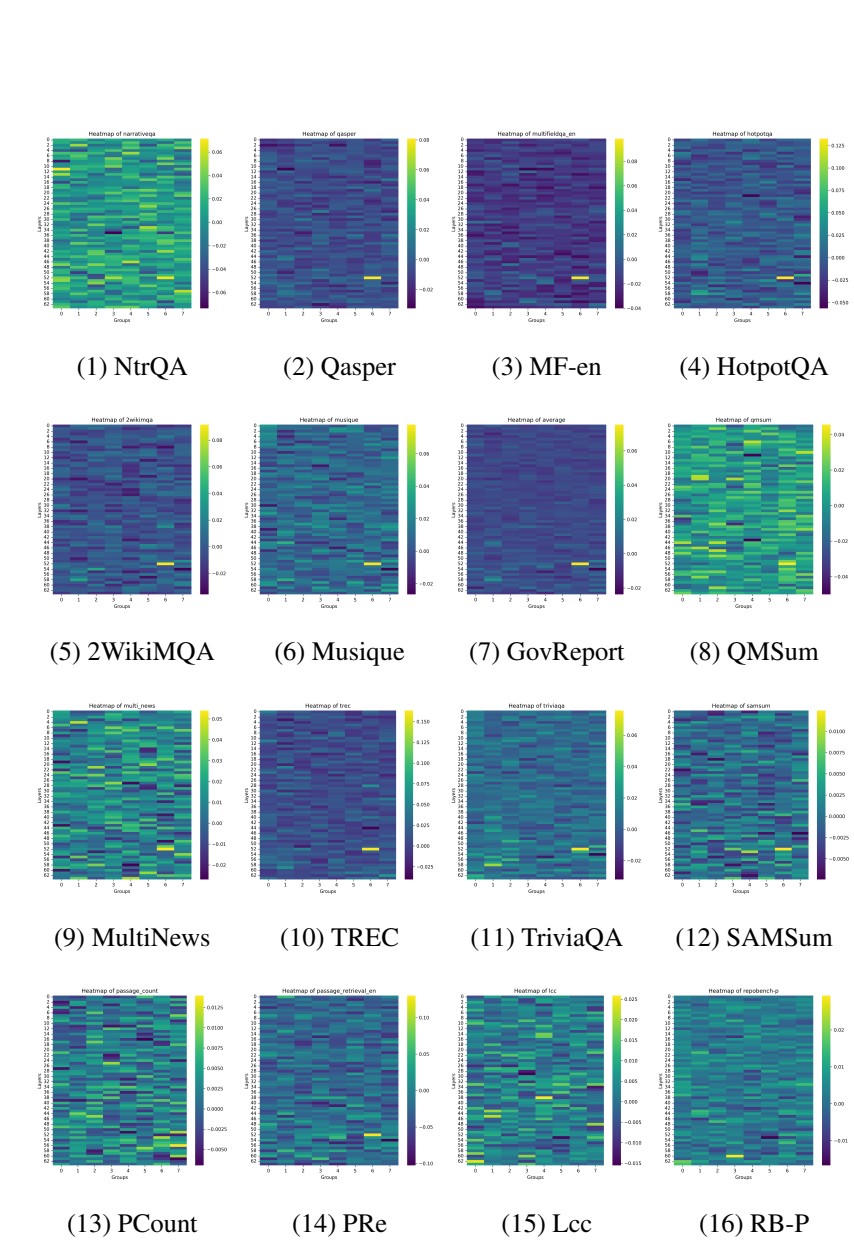

(1) NtrQA     (2) Qasper     (3) MF-en     (4) HotpotQA

(5) 2WikiMQA     (6) Musique     (7) GovReport     (8) QMSum

(9) MultiNews     (10) TREC     (11) TriviaQA     (12) SAMSum

(13) PCount     (14) PRe     (15) Lcc     (16) RB-P

Figure 6: Heatmap of Qwen3-32B.

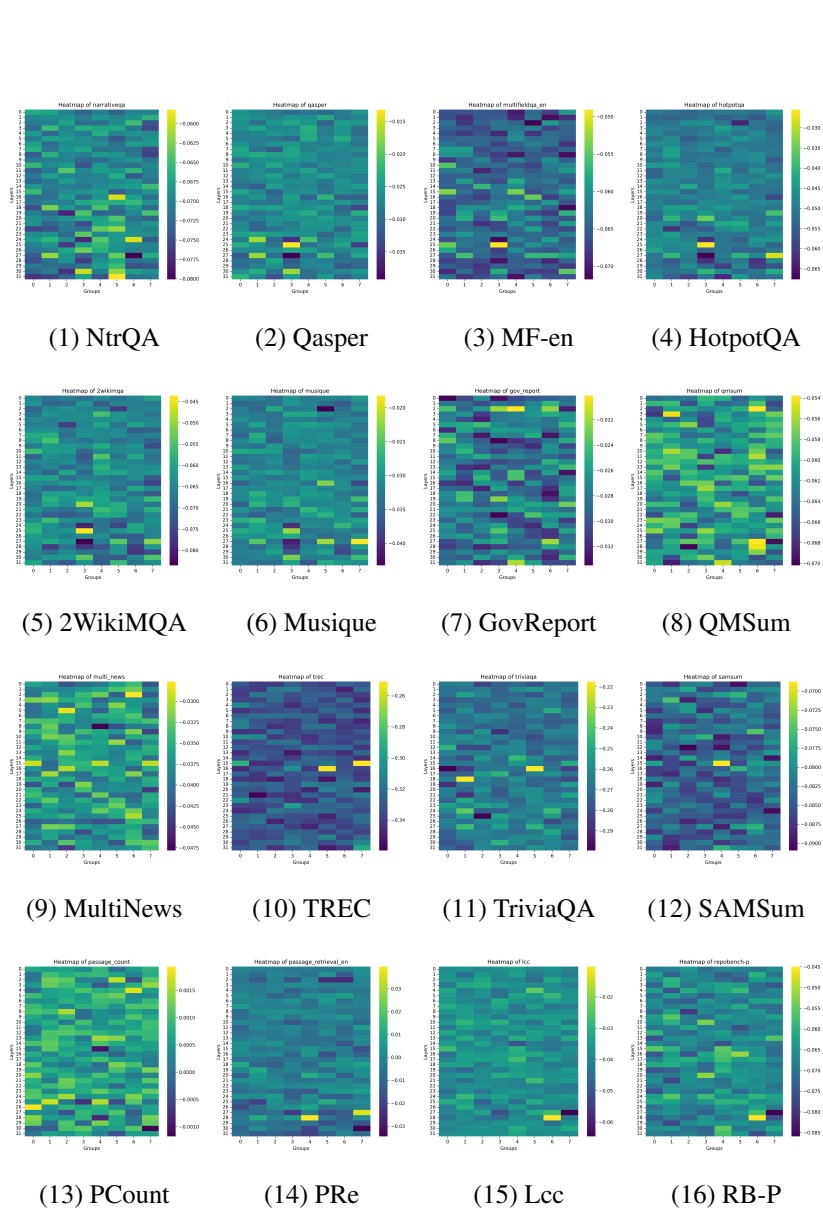

(1) NtrQA     (2) Qasper     (3) MF-en     (4) HotpotQA

(5) 2WikiMQA     (6) Musique     (7) GovReport     (8) QMSum

(9) MultiNews     (10) TREC     (11) TriviaQA     (12) SAMSum

(13) PCount     (14) PRe     (15) Lcc     (16) RB-P

Figure 7: Heatmap of Llama-3-8B-Instruct.

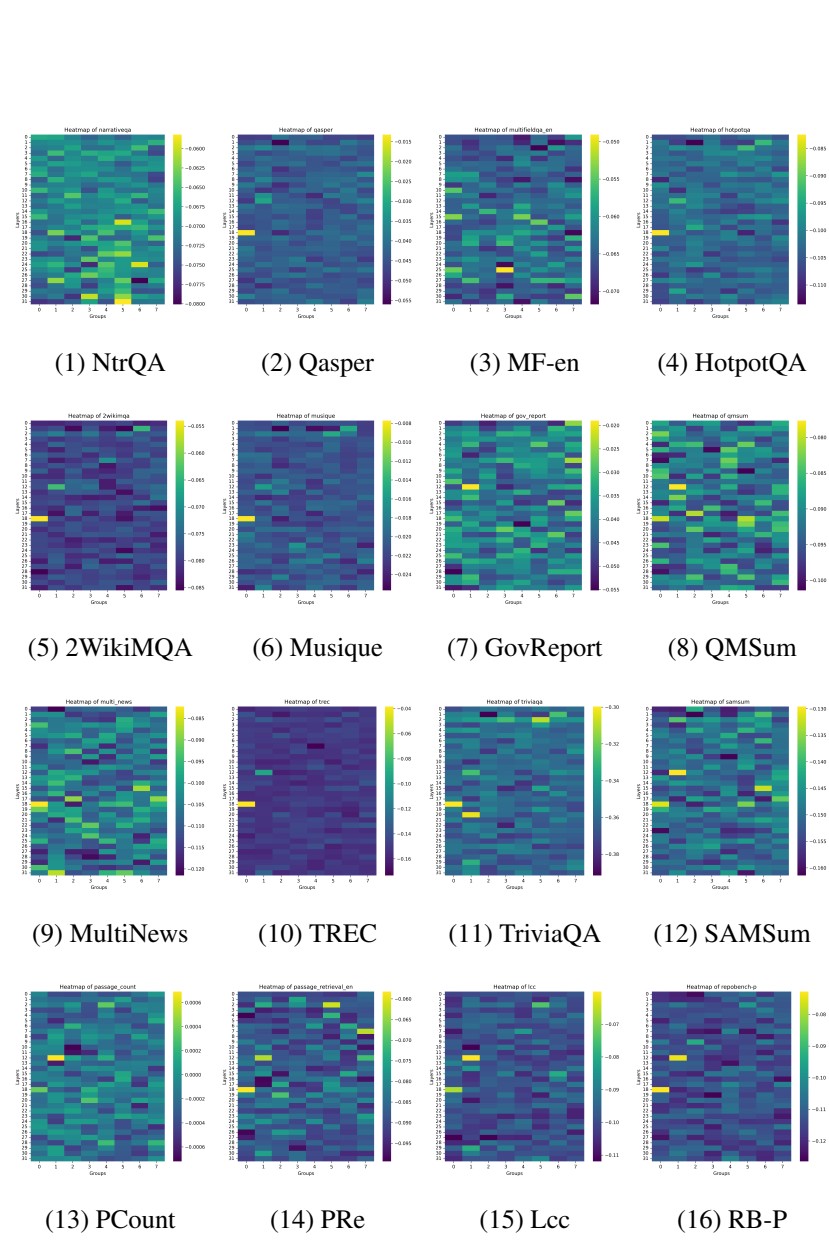

(1) NtrQA  (2) Qasper  (3) MF-en  (4) HotpotQA

(5) 2WikiMQA  (6) Musique  (7) GovReport  (8) QMSum

(9) MultiNews  (10) TREC  (11) TriviaQA  (12) SAMSum

(13) PCount  (14) PRe  (15) Lcc  (16) RB-P

Figure 8: Heatmap of Mistral-7B-Instruct-v0.2.

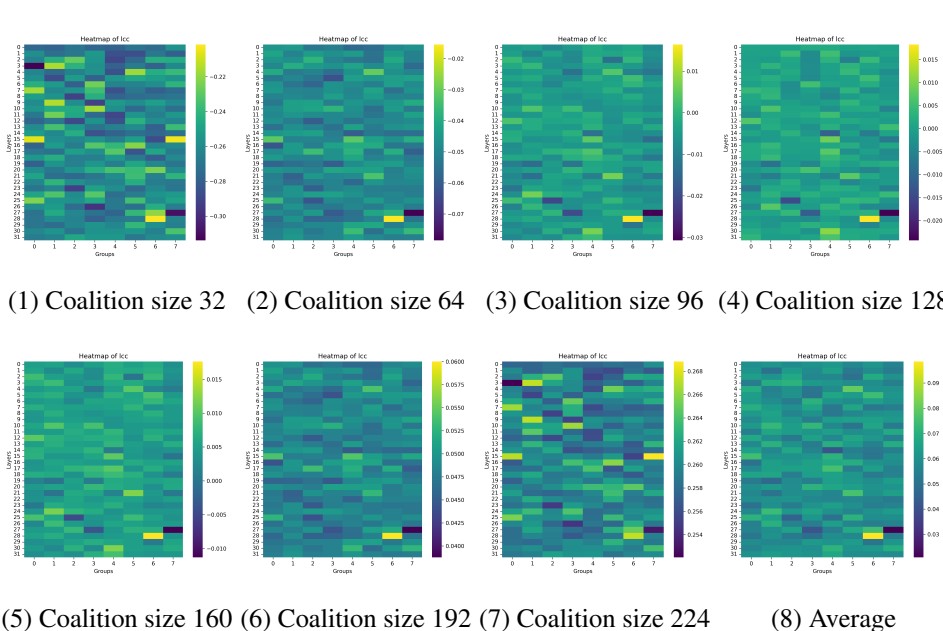

(1) Coalition size 32    (2) Coalition size 64    (3) Coalition size 96    (4) Coalition size 128

(5) Coalition size 160 (6) Coalition size 192 (7) Coalition size 224      (8) Average

Figure 9: The expected complementary contributions for the *lcc* dataset across different coalition sizes.

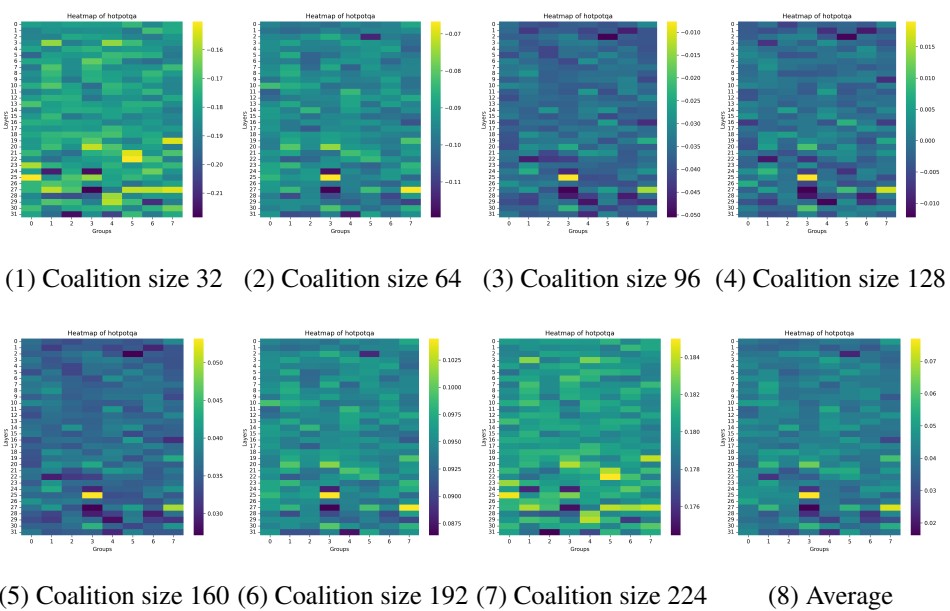

(1) Coalition size 32    (2) Coalition size 64    (3) Coalition size 96    (4) Coalition size 128

(5) Coalition size 160 (6) Coalition size 192 (7) Coalition size 224      (8) Average

Figure 10: The expected complementary contributions for the *hotpotqa* dataset across different coalition sizes.

Table 20: Generalization results of masking groups with Llama3-8B-Instruct

| Method | Multi-Doc. QA | | Code | |
|---|---|---|---|---|
| | 2WikiMQA | Musique | Lcc | RB-P |
| Full Cache | 34.56 | 21.09 | 58.10 | 51.64 |
| **Masking 16 groups** | | | | |
| Random | 20.01 | 20.6 | 49.83 | 40.55 |
| HeadKV-R2(top) | 17.33 | 14.32 | 26.62 | 26.53 |
| CoKV(top) | 10.78 | 5.43 | 14.41 | 15.33 |
| HeadKV-R2(low) | 27.23 | 12.55 | 37.35 | 38.55 |
| CoKV(low) | 39.92 | 20.9 | 64.04 | 61.22 |
| **Masking 32 groups** | | | | |
| Random | 18.50 | 6.94 | 30.78 | 40.71 |
| HeadKV-R2(top) | 9.37 | 5.11 | 11.24 | 13.64 |
| CoKV(top) | 6.71 | 3.45 | 4.39 | 5.78 |
| HeadKV-R2(low) | 14.48 | 7.42 | 33.13 | 32.39 |
| CoKV(low) | 38.1 | 18.22 | 64.75 | 58.28 |
| **Masking 64 groups** | | | | |
| Random | 9.11 | 6.76 | 21.27 | 18.07 |
| HeadKV-R2(top) | 12.01 | 2.46 | 21.03 | 16.14 |
| CoKV(top) | 5.68 | 1.82 | 2.5 | 3.66 |
| HeadKV-R2(low) | 14.21 | 6.63 | 16.00 | 14.58 |
| CoKV(low) | 34.17 | 16.29 | 49.97 | 48.93 |
| **Masking 96 groups** | | | | |
| Random | 10.33 | 5.08 | 15.37 | 9.81 |
| HeadKV-R2(top) | 9.14 | 2.93 | 13.55 | 14.76 |
| CoKV(top) | 4.38 | 1.28 | 2.74 | 3.07 |
| HeadKV-R2(low) | 16.80 | 8.00 | 10.56 | 11.05 |
| CoKV(low) | 28.08 | 12.92 | 38.62 | 40.55 |
| **Masking 128 groups** | | | | |
| Random | 5.12 | 2.73 | 12.27 | 9.23 |
| HeadKV-R2(top) | 7.19 | 1.85 | 13.94 | 11.76 |
| CoKV(top) | 2.93 | 0.94 | 2.48 | 3.84 |
| HeadKV-R2(low) | 12.83 | 5.26 | 10.14 | 6.03 |
| CoKV(low) | 24.34 | 9.37 | 23.38 | 24.11 |

Table 21: Generalization results of masking groups with Mistral-7B-v0.2

| Method | Multi-Doc. QA | | Code | |
|---|---|---|---|---|
| | 2WikiMQA | Musique | Lcc | RB-P |
| Full Cache | 26.07 | 17.81 | 55.10 | 49.45 |
| **Masking 16 groups** | | | | |
| Random | 27.19 | 17.83 | 52.38 | 48.24 |
| HeadKV-R2(top) | 22.76 | 14.29 | 39.49 | 35.66 |
| CoKV(top) | 13.02 | 6.99 | 17.97 | 23.38 |
| HeadKV-R2(low) | 24.88 | 16.93 | 54.22 | 49.4 |
| CoKV(low) | 26.25 | 18.18 | 54.58 | 50.03 |
| **Masking 32 groups** | | | | |
| Random | 22.55 | 11.92 | 50.14 | 47.18 |
| HeadKV-R2(top) | 20.31 | 12.86 | 33.41 | 30.47 |
| CoKV(top) | 10.23 | 5.16 | 11.8 | 13.64 |
| HeadKV-R2(low) | 23.55 | 13.28 | 52.89 | 48.85 |
| CoKV(low) | 26.61 | 17.62 | 55.35 | 49.92 |
| **Masking 64 groups** | | | | |
| Random | 16.93 | 15.65 | 38.07 | 28.39 |
| HeadKV-R2(top) | 13.79 | 8.07 | 26.25 | 21.23 |
| CoKV(top) | 4.52 | 2.11 | 13.14 | 13.31 |
| HeadKV-R2(low) | 14.26 | 7.41 | 38.78 | 39.22 |
| CoKV(low) | 33.11 | 16.97 | 52.68 | 49.54 |
| **Masking 96 groups** | | | | |
| Random | 10.41 | 8.43 | 21.30 | 17.65 |
| HeadKV-R2(top) | 14.22 | 3.99 | 17.73 | 16.24 |
| CoKV(top) | 2.09 | 3.04 | 10.96 | 8.32 |
| HeadKV-R2(low) | 10.10 | 5.44 | 34.95 | 32.57 |
| CoKV(low) | 31.51 | 17.39 | 47.71 | 45.37 |
| **Masking 128 groups** | | | | |
| Random | 7.80 | 2.07 | 23.74 | 18.34 |
| HeadKV-R2(top) | 6.15 | 3.89 | 18.68 | 18.52 |
| CoKV(top) | 1.19 | 3.42 | 9.81 | 6.0 |
| HeadKV-R2(low) | 7.20 | 6.64 | 24.09 | 24.48 |
| CoKV(low) | 23.76 | 12.12 | 42.01 | 36.7 |

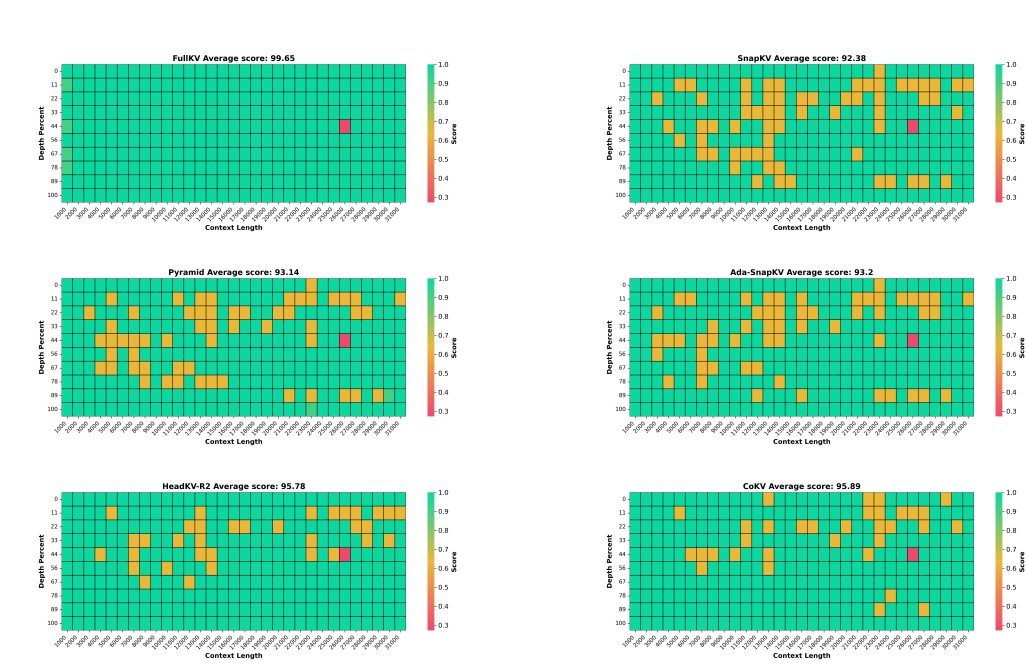

Figure 11: NIAH test results on Mistral-7B-v0.2 with average KV cache = 128

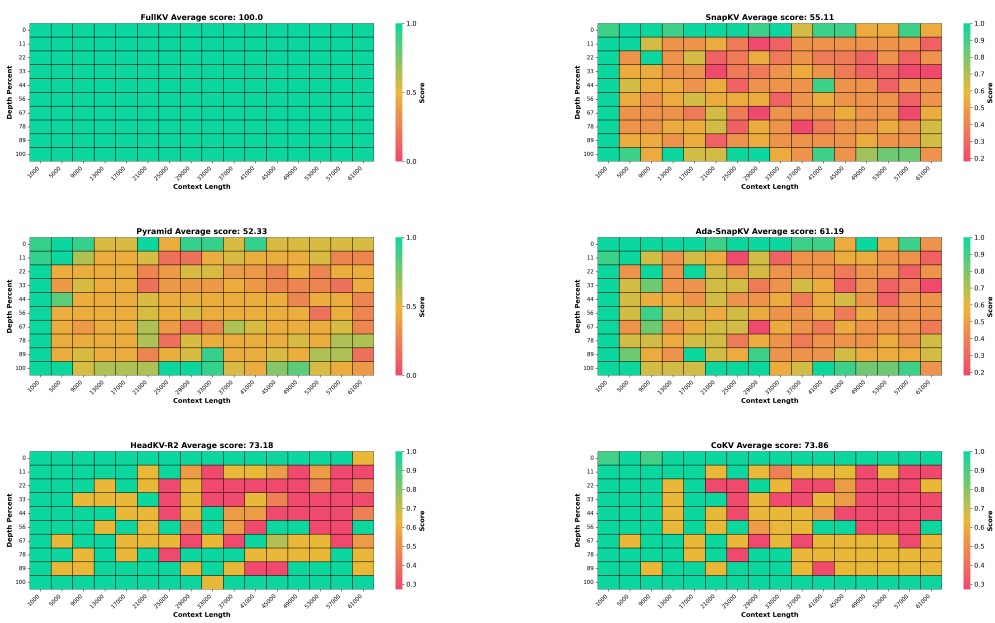

Figure 12: NIAH test results on Qwen3-32B with average KV cache = 128

Table 22: Evaluation results of masking 64 groups with Mistral-7B-Instruct-v0.2 across varying context lengths in the RULER benchmark

| Method | Retrieval | Multi-hop Tracing | Aggregation | Question Answering | |
|---|---|---|---|---|---|
| | niah | vt | fwe | qa1 | qa2 |
| **31k Context Length** | | | | | |
| Full Cache | 100.0 | 86.08 | 89.2 | 71.4 | 53.4 |
| Random | 0.8 | 59.6 | 54.4 | 53.6 | 19.4 |
| Headkv-R2(top) | 0.0 | 0.48 | 0.0 | 28.6 | 13.8 |
| CoKV(top) | 0.0 | 0.12 | 0.0 | 12.8 | 7.0 |
| Headkv-R2(low) | 92.4 | 67.08 | 76.73 | 41.8 | 33.0 |
| CoKV(low) | 100 | 90.88 | 74.33 | 70.2 | 51.6 |
| **16k Context Length** | | | | | |
| Full Cache | 100.0 | 90.44 | 94.73 | 76.8 | 54.6 |
| Random | 0.2 | 6.92 | 71.67 | 63.8 | 52.4 |
| Headkv-R2(top) | 0.0 | 0.56 | 0.0 | 29.2 | 17.0 |
| CoKV(top) | 0.0 | 0.76 | 0.27 | 15.4 | 7.2 |
| Headkv-R2(low) | 90.8 | 71.6 | 79.27 | 60.2 | 38.0 |
| CoKV(low) | 100.0 | 93.84 | 81.6 | 74.0 | 54.0 |
| **8k Context Length** | | | | | |
| Full Cache | 100.0 | 96.32 | 78.2 | 82.6 | 61.6 |
| Random | 99.4 | 64.6 | 58.8 | 71.2 | 51.0 |
| Headkv-R2(top) | 0.0 | 0.48 | 0.0 | 31.8 | 18.2 |
| CoKV(top) | 0.0 | 1.08 | 0.13 | 20.0 | 9.0 |
| Headkv-R2(low) | 91.2 | 69.4 | 61.4 | 64.2 | 44.0 |
| CoKV(low) | 100.0 | 92.72 | 67.13 | 80.4 | 62.0 |
| **4k Context Length** | | | | | |
| Full Cache | 100.0 | 99.32 | 84.6 | 85.0 | 63.0 |
| Random | 98.2 | 35.44 | 33.8 | 62.6 | 56.6 |
| Headkv-R2(top) | 0.0 | 0.16 | 0.0 | 30.4 | 20.0 |
| CoKV(top) | 0.0 | 1.16 | 0.27 | 20.8 | 8.4 |
| Headkv-R2(low) | 84.8 | 73.12 | 56.47 | 66.2 | 48.0 |
| CoKV(low) | 100.0 | 97.0 | 77.0 | 83.6 | 63.2 |

