# OpenReview forum: "CoKV: Optimizing LLM Inference with Game-Theoretic Adaptive KV Cache"
_ICLR.cc/2026/Conference — ICLR 2026 Conference Withdrawn Submission_

### Official Review · Reviewer_eVGh · 2025-10-17

**Soundness:** 3
**Presentation:** 2
**Contribution:** 2
**Rating:** 2
**Confidence:** 4

**Summary:**

This paper proposes CoKV, which models the cooperation among attention heads during inference as a cooperative game, allowing the analysis of each head's collaborative contribution to determine its importance. Based on this importance measure, the authors introduce a new cache eviction algorithm.

**Strengths:**

The idea of analyzing the cooperative relationships among attention heads appears reasonable and well-motivated.

**Weaknesses:**

1. The experimental setup, especially the training process of CoKV, is unclear. Specifically, the authors state:
"We randomly split each dataset into a very small validation dataset and a test dataset. The hyperparameter \alpha is selected from {1, 5, 10, 15, 20, 30, 40} based on the Sliced Shapley value computed on the corresponding validation set." However, it is not clear where the training set is. My understanding is that CoKV follows a training-based framework: there should be a training set used to learn the head scores, a validation set to select \alpha, and finally a test set to report the results. Did the authors train directly on the test set and then select \alpha using the validation set?

2. Following the previous point, it is also unclear how the authors report the LongBench scores. For example, in the Qasper dataset, there are about 200 samples. Were all 200 samples used to compute the final score?

3. CoKV seems to require training and hyperparameter tuning for each dataset individually. Does this approach have practical value in real-world applications?

4. The experimental design is somewhat confusing. The paper places significant emphasis on analyses based on masking important heads and comparing performance drops. In my view, such experiments serve more as analysis rather than direct evidence of the method's superiority. In contrast, the more detailed and valuable experiments-such as those on LongBench and Ruler-should be presented in the main text.

**Questions:**

The paper introduces several hyperparameters (e.g., \alpha, s, M). It would be helpful if the authors could provide a sensitivity analysis of these parameters and clarify the criteria for their selection in practical applications.

---

### Official Review · Reviewer_qWq7 · 2025-10-26

**Soundness:** 3
**Presentation:** 3
**Contribution:** 2
**Rating:** 6
**Confidence:** 3

**Summary:**

This paper proposes CoKV, a game-theoretic adaptive KV-cache optimization framework for large language model (LLM) inference.
The key insight is that prior KV-budget allocation methods treat each attention head independently, ignoring cooperative or redundant interactions between heads. CoKV models these relationships as a cooperative game, computing each head’s contribution using an approximate Sliced Shapley Value (SSV) to determine cache allocation for both KV eviction and quantization.

Experiments across Qwen-3-32B, Llama-3-8B-Instruct, and Mistral-7B on LongBench, RULER, GSM8K, and MATH show that CoKV achieves comparable or slightly better accuracy than HeadKV-R2 and Ada-KV while reducing peak memory by ≈ 38 % and inference latency to ≈ 25 % of the FullKV baseline.
The authors also report that CoKV can identify negative-contribution heads, improving interpretability and pruning safety.

**Strengths:**

1. Novel perspective:
Introduces a cooperative-game formulation of head importance—conceptually interesting and mathematically grounded.

2. Theoretical rigor:
Derives approximation bounds for the proposed SSV estimator, connecting sampling complexity with estimation variance.

3. Versatile applicability:
Works for both KV eviction and quantization, with only a single offline head-importance estimation step.

4. Empirical consistency:
Across diverse models and tasks, performance drop < 3 % while halving memory; decoding latency ≈ 25 % of FullKV.

5. Interpretability:
Ability to detect redundant or harmful heads offers a clear analytical advantage over heuristic-based approaches.

**Weaknesses:**

1. Limited performance gain:
Accuracy improvements are modest (≈ +1–2 points over HeadKV-R2 );
2. Offline cost:
The SSV estimation requires multiple inference passes on a validation set; although lightweight, it adds extra preprocessing.
3. Evaluation scope:
All experiments are head-level; token- or layer-level extensions are not demonstrated.
4. Ablation gaps:
Missing analysis on sensitivity to coalition size H and validation-set size; unclear scalability to 70B-class models.

**Questions:**

How does CoKV interact with FlashAttention 2 and PagedAttention caching mechanisms?

---

### Official Review · Reviewer_61vm · 2025-10-28

**Soundness:** 2
**Presentation:** 3
**Contribution:** 2
**Rating:** 2
**Confidence:** 4

**Summary:**

This paper introduces a cooperative game-theoretic approach for modeling interactions among attention heads in large language models. By quantifying each head's collaborative contribution, CoKV enables more accurate and efficient KV cache budget allocation, including eviction and quantization. Experiments on Longbench validate its effectiveness.

**Strengths:**

1. The paper is well-organized and easy to follow.

2. The results on LongBench demonstrate that CoKV can more effectively identify the importance of attention heads.

**Weaknesses:**

1. The CoKV method searches for hyper-parameters separately on each dataset but does not specify the selected values or provide any robustness analysis.

2. The training process of CoKV introduces substantial computational overhead, especially since it requires grid search over different hyper-parameters.

3. As shown in Figure 6, the training outcomes of CoKV vary significantly across different datasets, which may limit its practicality for real-world deployment.

4. The experiments are limited to evaluations on LongBench and should include additional benchmarks such as RULER and mathematical reasoning datasets for a more comprehensive assessment.

**Questions:**

n/a

---

### Official Review · Reviewer_P5da · 2025-10-31

**Soundness:** 2
**Presentation:** 3
**Contribution:** 2
**Rating:** 4
**Confidence:** 5

**Summary:**

This paper propose the adaptive method to allocate the budget for different attention heads by using the cooperative contributions within the model. Extensive experiments demonstrate the effectiveness of CoKV on long-context benchmarks (e.g., LongBench, NIAH, and RULER) and mathematical reasoning benchmarks (e.g., GSM8K and MATH) across multiple model families, including Qwen, Llama, and Mistral.

**Strengths:**

The adaptive budget allocation is useful by using the cooperative contributions within the model, the head importance is measured using the Sliced Shapley Value (SSV) with the complementary contributions. Then the KV Cache is compressed with adaptive budget allocation, Heads with higher SSV are allocated more cache size or bit width to retain KV pairs prior to the local window.

The experiments show good results: For Qwen3-32B, CoKV achieve 98.83% of the performance of the full KV when retains an average of 1024KV pairs(12.8%). Also Experiments on mathematical reasoning datasets also demonstrate that CoKV possesses strong cross-task capabilities.

**Weaknesses:**

A lot of works, just like AdaKV, the adaptive budget allocation is very mature. Maybe the Attention recall based method is enough, so this should give the theoretical analysis of the comparison between the proposed method and Attention recall.

And the experiments on long-cot tasks is not enough, since the performance of reasoning is very sensitive to token eviction.

**Questions:**

1. What is the advantage of this method, compared with the method based Attention recall?
2. In the future, the native sparse attention may be popular, how can your method adapt to these methods?
3. How this method adapted to PageAttention, for system implementation?
4. How can this method used with quantization?

---

### Note · Authors · 2025-11-12

I have read and agree with the venue's withdrawal policy on behalf of myself and my co-authors.